https://doi.org/10.1038/s41467-019-13996-4　　**OPEN**

# Word contexts enhance the neural representation of individual letters in early visual cortex

Micha Heilbron[1,2]*, David Richter [1], Matthias Ekman [1], Peter Hagoort [1,2] & Floris P. de Lange[1]

Visual context facilitates perception, but how this is neurally implemented remains unclear. One example of contextual facilitation is found in reading, where letters are more easily identified when embedded in a word. Bottom-up models explain this word advantage as a post-perceptual decision bias, while top-down models propose that word contexts enhance perception itself. Here, we arbitrate between these accounts by presenting words and non-words and probing the representational fidelity of individual letters using functional magnetic resonance imaging. In line with top-down models, we find that word contexts enhance letter representations in early visual cortex. Moreover, we observe increased coupling between letter information in visual cortex and brain activity in key areas of the reading network, suggesting these areas may be the source of the enhancement. Our results provide evidence for top-down representational enhancement in word recognition, demonstrating that word contexts can modulate perceptual processing already at the earliest visual regions.

[1] Donders Institute for Brain, Cognition and Behaviour, Radboud University, NL-6500 HBNijmegen, The Netherlands. [2] Max Planck Institute for Psycholinguistics, 6525 XD Nijmegen, The Netherlands. *email: m.heilbron@donders.ru.nl

Context-based expectations can strongly facilitate perception, but how this is neurally implemented remains a topic of debate[1,2]. One famous and striking example of contextual facilitation is found in reading, where letters are more easily identified when embedded in a linguistic context such as a word or name (e.g., a road sign) than in a random string (e.g., a license plate)[3].

Historically, two opposing accounts have been proposed to explain this so called 'word superiority effect'. Under the guessing-based account, letter identification occurs in a bottom-up fashion and the advantage offered by words constitutes only a post-perceptual advantage in 'guessing' the correct letter[4,5]. Alternatively, the perceptual account explains word superiority as a top–down effect, proposing that higher-order linguistic knowledge can enhance perceptual processing of the individual letters[6,7]. A rich behavioural literature, dating back several decades[8,9], has documented that even when the ability to guess the correct letter is experimentally controlled, the word advantage persists[10]. This has been interpreted as evidence that the effect must (at least in part) reflect top–down perceptual enhancement—a view that remains dominant until today[11].

However, some lingering doubts have persisted. For instance, ideal observer analysis has shown that the efficiency of letter recognition is much lower than that of a fully holistic (word-based) observer, and lies within the theoretical limits of a strictly letter-based (feedforward) observer—even when considering word superiority[12]. Moreover, advances in deep learning have shown that letters and other complex objects can be accurately recognised in context by bottom-up architectures, further questioning the need to invoke top–down explanations[13]. Beyond these theoretical arguments, neural evidence for the perceptual locus of this supposedly top–down effect is lacking. This is remarkable, since the top–down interpretation of word superiority makes a clear neural prediction: if the behavioural word advantage is due to a perceptual enhancement of letter stimuli, then it should be accompanied by an enhancement of sensory information in the early visual areas that process the individual letters already.

Here, we test this prediction using a simple paradigm involving streams of words and nonwords. We use neural network simulations of the paradigm to confirm that top–down models would uniquely predict the enhancement of letter representations by word contexts. When we then perform the same experiment in human observers while recording brain responses using functional magnetic resonance imaging (fMRI), we find that word contexts robustly enhance letter representations in early visual cortex. Moreover, compared to nonwords, words are associated with increased information-activation coupling between letter information in early visual cortex on the one hand, and blood-oxygen-level-dependent (BOLD) activity in key areas of the reading network on the other. These results suggest word superiority is (at a least in part) a perceptual effect, supporting prominent top–down models of word recognition.

## Results

### Word contexts facilitate orthographic decisions.
Participants ($n = 34$) were presented with streams of words or nonwords consisting of five letters (see Fig. 1a), while maintaining fixation. We used a blocked design in which word and nonword (i.e. unpronounceable letter string) stimuli were presented in long trials of ten items of which the middle letter (U or N) was kept fixed while the outer letters varied, creating a word or nonword context (each 10-s trial containing only stimuli of one condition). To make reading visually challenging, stimuli were embedded in Gaussian noise (see Methods). To keep participants engaged, they

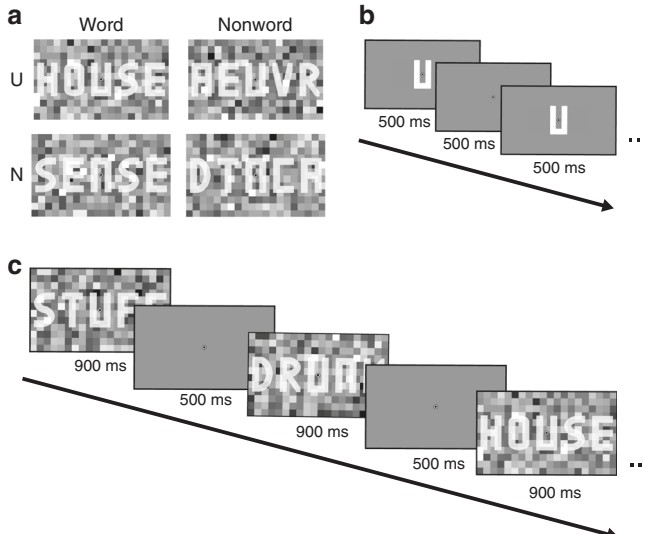

**Fig. 1 Experimental paradigm. a** Example stimuli for each condition. Participants observed words or nonwords (i.e. orthographically illegal, unpronounceable strings) with a U or N as middle letter, resulting in four conditions. **b** Functional localiser. During the functional localiser, the key letters (U and N) were presented in isolation and without visual noise, while participants performed an irrelevant task at fixation. **c** Trial structure. We used a blocked design, in which each 14-s trial consisted of ten words or nonwords with a fixed middle letter. Participants performed an orthographic discrimination task on specific, prelearned targets that occurred once or occasionally twice per trial. Participants were trained in a separate session to perform the task while maintaining fixation at the centre of the screen.

performed a spelling discrimination task on specific target stimuli that occurred occasionally (1–2 times) per trial. Target stimuli were learned during a prior training session. Targets were presented either in their regular form or with one letter permuted, and participants had to categorise targets as 'spelled' correctly or incorrectly (i.e. presented in the learned form or permuted). Participants were faster (median RT difference: $-29.2$ ms; Wilcoxon signed rank, $T_{34} = 40$, $p = 1.07 \times 10^{-5}$, $r = 0.87$) but not significantly more accurate (mean accuracy difference: 1.62%; $t$-test, $t_{34} = 1.70$, $p = 0.098$, $d = 0.29$) for word compared to nonword targets. This observation is in line with the word superiority effect, but from the behaviour alone it is unclear whether the word advantage was perceptual or post-perceptual.

### Representational enhancement is a hallmark of top–down models.
Because our paradigm is different from the traditional paradigms in the (behavioural) word superiority literature, we performed simulations of our experiment to confirm that the top–down account indeed predicts the representational enhancement we set out to detect. We used a predictive coding implementation[14] of the influential Interactive Activation architecture proposed by McClelland and Rumelhart[6] (see Methods).

In the simulation, we ran artificial 'runs' in which we presented sets of word and nonword stimuli used in the experiment to the network (Fig. 2a). To simulate experimental viewing conditions, we added Gaussian noise and ran the network until convergence so as to mimic long stimulus duration (see Methods) resulting in stimuli that were presented well-above recognition threshold (Supplementary Fig. 2). Representational strength was quantified by dividing the activity level for the correct letter unit by the sum of activity levels of all letters—a fraction that asymptotically goes to 1 as representational strength increases. After running 34 simulated runs with the top–down model, the relative evidence

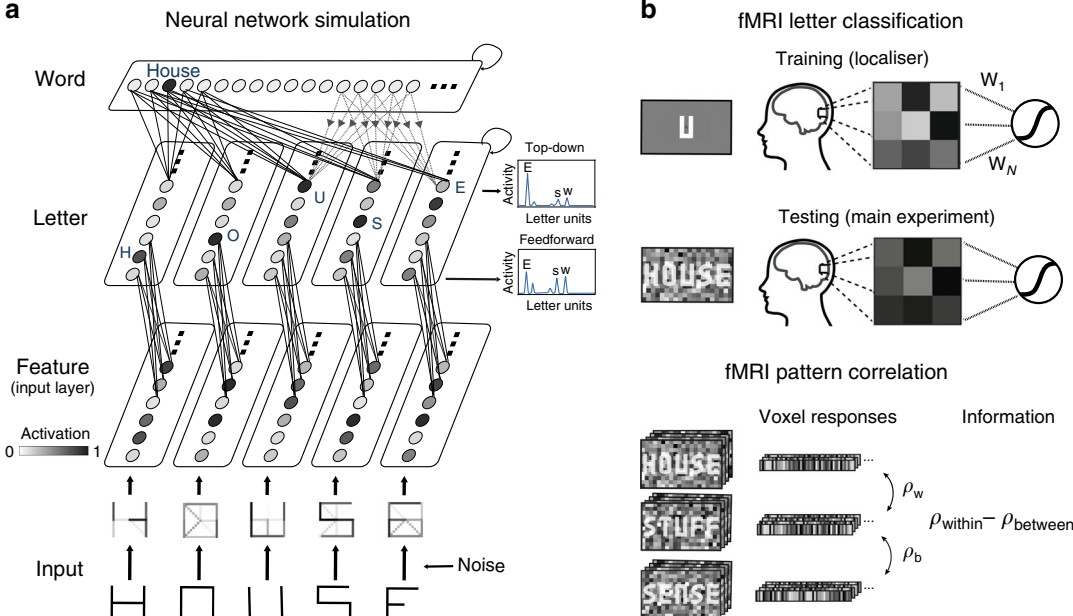

**Fig. 2 Probing representational enhancement in neural network models and the brain. a** Modelling representational enhancement in a hierarchical neural network model[6,14]. Stimuli used in the experiment were encoded into vectors of visual features and overlaid with Gaussian noise (bottom rows). Inputs were presented to a network with or without word-to-letter feedback connections. For both networks, representational strength was quantified from the distribution of activity levels of letter units for the third position (principle illustrated for the fifth letter, E). Solid circles indicate units (representing features, letters or words); lines indicate feedforward connections, and dotted lines with arrows indicate feedback connections. Note that we used a predictive coding formulation of the network[14] but for simplicity only the state estimator (prediction) units are shown in the schematic (see Methods for details). **b** Quantifying representational letter enhancement using multivariate pattern analysis (MVPA). To probe letter representations in the brain, we used two MVPA techniques: classification (upper panel) and pattern correlation (lower panel).

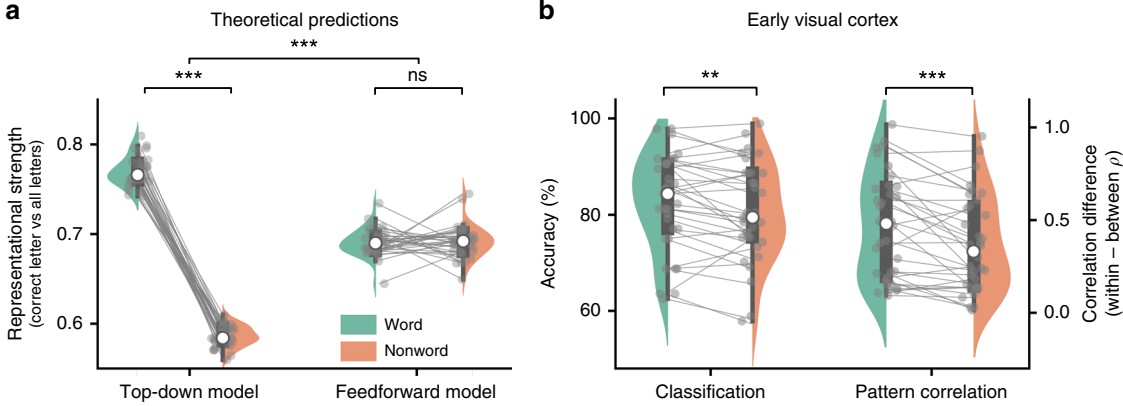

**Fig. 3 Word contexts enhance letter representations. a** Theoretical predictions. We simulated 34 artificial 'runs' in which we exposed a network with (top–down model) and without (feedforward model) word-to-letter feedback connections to the experimental stimuli, and computed the average representational strength of the middle letter in word and nonword contexts. Note that the strong dissociation was observed despite the fact that the middle letter was well-above threshold in all conditions for both models (Supplementary Fig. 2). **b** Letter representation in early visual cortex of 34 human observers. Two multivariate pattern analysis methods (see Methods) revealed that neural representations of letters were enhanced in word compared to nonword contexts, supporting the top–down model. In both panels, grey dots represent individual simulated 'runs' (**a**) or individual participants (**b**). Lines represent paired differences. White dots, boxes and whiskers represent between-subject medians, quartiles and 1.5 interquartile ranges, respectively. Significance levels correspond to $p < 0.01$ (**) or $p < 0.001$ (***) in a paired, two-tailed Student's $t$ or Wilcoxon sign rank test.

for the middle letter was confirmed to be much higher in words than nonwords (paired $t$-test, $t_{34} = 50.5$, $p = 7.72 \times 10^{-33}$), despite the signal-to-noise ratio of the simulated stimuli being identical (Fig. 3a). Importantly, when the same stimuli were presented to a network lacking word-to-letter feedback connections, no such difference was found (paired $t$-test, $t_{34} = -0.24$, $p = 0.81$), resulting in a significant interaction (two-sample $t$-test $t_{34} = 31.1$, $p = 3.5 \times 10^{-41}$). This confirmed that despite the

differences between our and the classic paradigm, representational enhancement of letters by word contexts is a hallmark of top–down models of letter perception.

**Word contexts enhance letter representations.** Next, we tested whether we could find a similar enhancement effect in early visual cortex in our participants. To do so, we first trained a classifier for

each participant on an independent dataset from a functional localiser run, during which the two middle letters (U or N) were presented in isolation and without Gaussian noise (see Fig. 2b). We then tested the classifier's ability to identify the middle letter of the words and non-words presented in the main experiment, in a trial-based fashion (each trial lasting 14 s and consisting of ten stimuli). We reasoned that if word context enhances the sensory representations of letters (e.g. enhancing the letter features in noise), this should be apparent in early visual areas, which we defined as the union of V1 and V2 (see Methods). To focus on voxels sensitive to the relevant part of the visual field, we selected the 200 voxels (the same number we used in a previous study[15]) most responsive during the localiser run. We were able to classify letter identity well above chance level (one sample t-test, $t_{34} = 18.84$, $p = 3.13 \times 10^{-19}$, $d = 3.23$) reaching a mean overall decoding accuracy of 81.4% averaged over both conditions (see Fig. 3).

Having established that letter identity can be extracted with high fidelity from early visual cortex, we went on to test if representational content was enhanced by word context. Strikingly, we found that classification accuracy was indeed higher for words compared to nonwords (Wilcoxon sign rank test, $T_{34} = 141.5$, $p = 7.55 \times 10^{-3}$, $r = 0.52$; Fig. 3b). To further examine this enhancement effect, we quantified representational content using an (arguably simpler) supplementary multi-voxel pattern analysis (MVPA) technique: pattern correlation analysis—the difference in voxel response pattern correlation that could be attributed to letter identity ('Pearson $\rho$ within-letter' minus 'Pearson $\rho$ between-letter'; see Methods). Reassuringly, the results aligned with those of the classification analysis: the correlation difference score being significantly higher for words than nonwords (Wilcoxon sign rank, $T_{34} = 103$, $p = 8.83 \times 10^{-4}$, $r = 0.67$).

To confirm that the differences revealed by the classification and pattern correlation analysis were related to differences in representations of stimulus information and not to unrelated confounding factors, we performed a number of controls. First, we tested the stability of the results over different region of interest (ROI) definitions. Since both representational analyses used the 200 voxels that were most responsive during an independent functional localiser, we wished to ensure that the results were not unique to this a priori specified (but arbitrary) number. We therefore re-ran the same analyses for ROIs ranging from 50 to 1000 voxels with steps of 10. This revealed that the same pattern of effects was found over practically the entire range of ROI sizes (Supplementary Fig. 3).

Another possibility is that the increased estimates of representational content could be explained by a simple difference in signal amplitude, potentially related to participants being more attentive to words than nonwords. To address this, we quantified BOLD amplitude per condition using a standard generalised linear model (GLM) based approach (see Methods), but found no significant difference between conditions in the amplitude estimates for the corresponding voxels (paired t-test, $t_{34} = -0.57$, $p = 0.57$, $d = 0.10$; Bayesian paired t-test, $BF_{10} = 0.21$; see Supplementary Fig. 4). Importantly, we found no significant differences in eye-movement deviation from fixation between words and nonwords (Wilcoxon $T_{32} = 197$, $p = 0.21$, $r = 0.25$; Bayesian paired t-test, $BF_{10} = 0.48$; see Supplementary Fig. 6 and Methods), confirming participants' ability to maintain fixation during the task did not differ significantly between conditions.

As a final control analysis, we wanted to confirm that the MVPA results relied on retinotopically specific information. This would be an important indication that both the letter information extracted from visual cortex, and its enhancement by word contexts, indeed originate from sensory representations. To this end, we performed a searchlight variant of the classification and pattern correlation analysis (see Supplement for details). This

revealed (see Supplementary Figs. 7, 8) that letter identity information was only visible in neural activity patterns in visual cortex, ruling out that decoding relied on a brain-wide signal. We further tested for retinotopic specificity within visual cortex by comparing the functionally defined central ROI (described above), to a functionally defined peripheral ROI (see Methods and Supplementary Note 1 for more details). This revealed (Supplementary Fig. 9) that overall letter decoding was greatly reduced for the peripheral ROI compared to the central ROI, both for classification (paired t-test, $t_{34} = 15.59$, $p = 8.86 \times 10^{-17}$, $d = 2.67$) and pattern correlation analysis (paired t-test, $t_{34} = 8.06$, $p = 2.65 \times 10^{-9}$, $d = 1.38$). Importantly, we found a similar reduction in the peripheral ROI for the enhancement effect (the difference in decoding between conditions), again both for the classification (paired t-test, $t_{34} = 2.56$, $p = 0.015$, $d = 0.44$) and pattern correlation analysis (paired t-test, $t_{34} = 2.92$, $p = 6.31 \times 10^{-3}$, $d = 0.50$).

In sum, these analyses show that sensory letter information in early visual cortex, as estimated by classification and pattern correlation analysis, was increased in words compared to nonwords. This enhancement was present over a range of ROI definitions, but was reduced for peripheral compared to central ROIs, and could not be explained by confounding factors such as BOLD amplitude or eye movements.

**Representational enhancement across the visual hierarchy**. Having established a perceptual enhancement effect by word context in early visual cortex, we then asked how this enhancement effect was distributed among specific visual areas. To this end, we further investigated five ROIs, four of which were defined anatomically (V1–V4) and one (visual wordform area (VWFA)) functionally; in each ROI, voxels were selected using the procedure described earlier (see Methods for details).

The results show consistent evidence for word enhancement in V1, V2 and V4 (all $p$'s < 0.025; see Supplementary Fig. 10 for details), with both analyses. In contrast, V3 and VWFA showed no consistent evidence for word enhancement (see Supplementary Fig. 10). However, in these regions, the overall classification accuracy and pattern information scores were also close to chance, making the absence of differences between conditions difficult to interpret. For regions V1–V4, we also tested for univariate amplitude differences between word and nonword conditions. Interestingly, in all four regions, the sign of the univariate difference was negative (indicating weaker amplitude of responses to word stimuli), but note that only in V4 this difference was marginally significant (paired t-test, $t_{34} = -2.11$, $p = 0.04$, $d = -0.36$, uncorrected; Supplementary Fig. 5). In sum, we observed word enhancement across multiple regions in the visual hierarchy. Critically, none of the regions showed BOLD amplitude differences, ruling out the possibility that word enhancement was confounded by low-level attentional differences between conditions.

**Information-activation coupling reveals putative neural sources**. Having observed a hallmark of top–down perceptual enhancement by word contexts, we then asked what the potential neural source of this top–down effect could be. We reasoned that if a candidate brain region was involved in the observed enhancement, then activity levels in this region would be expected to covary with the amount of letter information represented in early visual cortex. Moreover, this relationship should not be driven by a categorical difference between conditions (e.g. that both BOLD amplitude in a candidate region and informational content in visual cortex are higher for words than nonwords, while the two are not related within conditions). Taking the two

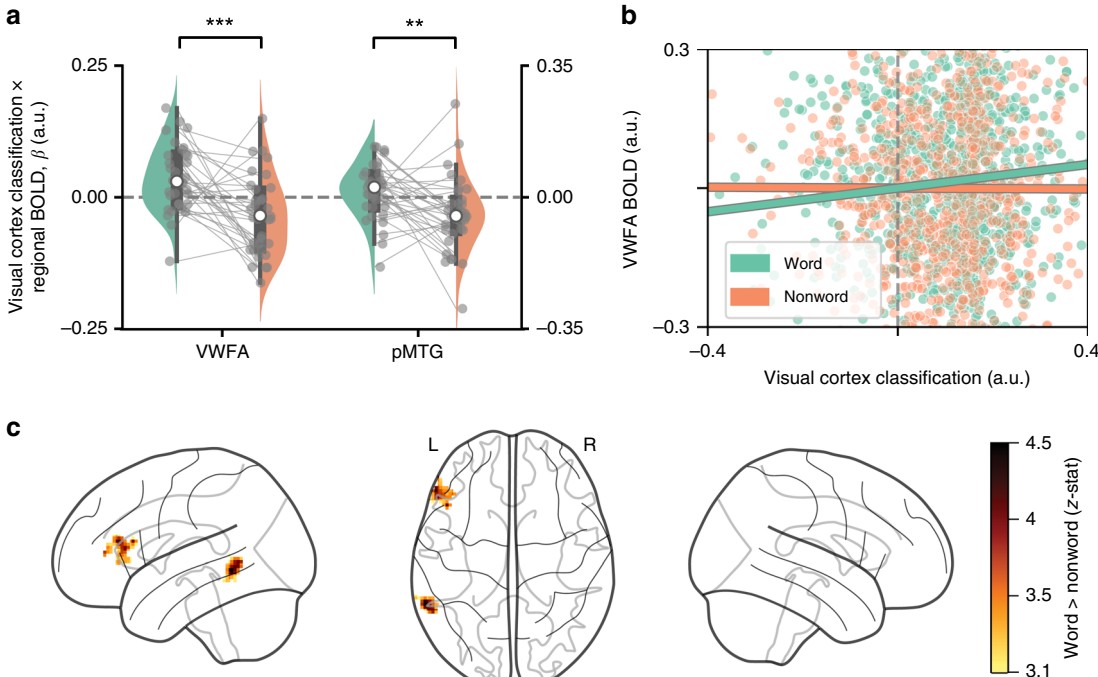

**Fig. 4 Information-activation coupling analysis. a** ROI-based coupling analysis. For two ROIs, GLMs were fitted to estimate coupling between early visual cortex classification evidence and regional BOLD amplitude for words and nonwords separately. We then tested for increased coupling (higher coefficients) in words compared to nonwords. **b** Illustration for example participant. A single (averaged) timecourse was extracted from each ROI and regressed against visual cortex classification evidence to test for increased slopes in words compared to nonwords. For illustration purposes, only the predicted slopes based on the regressor of interest are shown. Note that classification evidence was defined as probability, but here expressed in arbitrary units due to the whitening operation. **c** Whole-brain results. Same analysis as in **a** and visualised in **b**, but performed for each voxel independently. Resulting contrast images (word-nonword) were tested at the group level for increases in coupling in words compared to nonwords. This revealed statistically significant clusters ($p < 0.05$ FWE corrected), in the left pMTG and in the left IFG. Glass brain plot rendered with nilearn[63]. For a non-thresholded slice-by-slice rendering of the whole-brain results in **c**, see Supplementary Fig. 11. Grey dots indicate coefficients of individual participants, and lines the within-subject differences; white dots, boxes and whiskers are between-subject medians, quartiles and interquartile ranges, respectively. Significance stars correspond to $p < 0.01$ (**) or $p < 0.001$ (***) in a paired two-tailed $t$-test or Wilcoxon sign rank test. pMTG posterior medial temporal gyrus, IFG inferior frontal gyrus.

requirements together, we expected regions implicated in the top–down effect to show increased functional coupling between local BOLD activity and representational information in early visual cortex, for words compared to nonwords.

To test for this increased information-activation coupling, we used a GLM-based approach to model regional BOLD amplitude in both conditions as a function of early visual cortex classification evidence, and tested for an increased slope for words compared to nonwords (see Fig. 4b). This is analogous to the well-established psychophysiological interaction (PPI) analysis[16], but uses classifier evidence instead of BOLD activity as seed timecourse. Classifier evidence here corresponds to the predicted probability of the correct (presented) letter stimulus for each brain volume (see Methods).

We first tested for increased coupling in a hypothesis-driven, ROI-based fashion. We tested two candidate regions: the VWFA and the left posterior middle temporal gyrus (pMTG), associated with orthographic/visual[17,18] and lexical/semantic processing[19,20], respectively. Activity of all voxels was averaged to obtain a single BOLD timecourse per ROI. This BOLD timecourse was then modelled as a function of visual cortex classification strength to obtain separate coupling parameters for word and nonword conditions. We indeed observed a significantly increased coupling in both VWFA (Wilcoxon sign rank, $T_{34} = 80$, $p = 2.00 \times 10^{-4}$, $r = 0.73$) and pMTG (paired $t$-test $t_{34} = 2.83$, $p = 8.2 \times 10^{-3}$, $d = 0.48$; see Fig. 4a) for words. The increase in coupling appeared stronger in VWFA, but the difference in effects between regions was not statistically significant (paired $t$-test, $t_{34} = 0.62$, $p = 0.54$, $d = 0.11$).

Finally, we carried out an exploratory analysis by testing for increased functional coupling across the entire brain. In essence, the GLM procedure was identical to the one above, but carried out at the individual voxel level. This yielded, for each participant, a map of estimated differences in functional coupling for every voxel. These functional maps were then registered to a standard space after which we tested whether there were clusters of voxels that showed an increase in functional coupling for words compared to nonwords. We found two significant (FWE-corrected, cluster-forming $p < 0.001$, cluster-level $p < 0.05$) left-lateralised clusters at key nodes of the language network: one in pMTG and one in inferior frontal gyrus (IFG) (Fig. 4c; Supplementary Fig. 11). No significant cluster was found at VWFA, possibly due to individual neuro-anatomical variability in VWFA size and location[21].

Altogether, these results demonstrate increased functional coupling between visual cortical classification evidence and neural activity in VWFA, pMTG and IFG. In all of these regions, we found a significant increase in functional coupling (here, meaning that classifier evidence increased when the regions became more active, and vice versa) for words compared to nonwords, which is consistent with the idea that these regions might constitute the neural source of the top–down effect.

## Discussion

Visual context facilitates perception[1]. Letter perception offers a striking example of such facilitation, as letters are more easily

recognised when embedded in a word. Dominant, 'interactive' models of word recognition assume this facilitation occurs in the visual system already, proposing that linguistic knowledge can enhance perception in a top–down fashion[6]. Here we tested this perceptual enhancement hypothesis at the neural level. We presented streams of words or nonwords with a fixed middle letter while recording fMRI. Simulations of this paradigm confirmed that top–down models of word recognition uniquely predict that perceptual representations of the middle letter should be enhanced when embedded in a word. In line with the top–down account, information about the middle letter, probed using MVPA in early visual cortex, was enhanced when the letter was embedded in words compared to nonwords. Further, we found increased functional coupling between the informational pattern in early visual cortex, and regional BOLD amplitude in three key regions of the left-lateralized language network, i.e. VWFA, left pMTG and IFG. This points to these regions as potential neural sources of the representational enhancement effect. Together, these results constitute the first neural evidence for representational enhancement of letters by word contexts, as hypothesised by top–down accounts of word recognition[6]. The results also fit naturally with theoretical frameworks of top–down perceptual inference, such as hierarchical predictive coding[22–24] (see refs. [2,25] for review) and with the broader literature on top–down, predictive effects in language processing[26–28].

Our results are in line with a large behavioural literature on context effects in letter perception that support interactive activation (top–down) models. These works have demonstrated, for instance, that the word advantage persists when the guessing advantage afforded by words is constrained experimentally[8,9], see ref. [10] for review; that readers subjectively perceive letters embedded in real words as sharper[29]; and that readers are better at detecting subtle perceptual changes in real words than in nonwords[29]. While the behavioural literature has extensively investigated top–down effects, work on the neural basis of visual word recognition has focussed almost exclusively on its bottom-up component, most notably by probing the bottom-up selectivity profile of VWFA (or ventral occipitotemporal cortex more broadly) to various visual and orthographic properties[30–32]. One study tried to disentangle word and letter encoding at a neural level[33], but did not probe individual letter representations and their enhancement by word contexts. A recent study did investigate top–down processing, but was limited to attention-based response modulations in decision contexts[34].

Beyond the domain of language, but converging with the results presented here, are results from object perception, where it was recently found that the facilitation of object recognition by familiar contexts was accompanied by enhancement of object representations in object selective cortex[35]. The similarity to the current findings speaks to the idea that representational enhancement reflects a more general principle of contextual effects in perception. These contextual effects have been extensively studied, and range from neurons in macaque early visual cortex responding differently to identical lines when presented as parts of different figures[36,37] to neurons in mouse V1 encoding contextually expected but omitted stimuli (see ref. [38] for review). As such, it may be that although here the context is linguistic, the contextual effect in visual cortex reflects a more general mechanism that is not specific to reading or unique to humans. Interestingly, the idea that contextual enhancement reflects a more general perceptual mechanism was a key motivation to develop models of word recognition in the first place[6,7].

When viewed as a more general principle of perception, contextual enhancement touches on an even broader question: are objects recognised by their parts or as wholes? On the one hand, word superiority has historically been taken as an example of 'holistic' perception[8,10] and the enhancement we observed (in which 'wholes' enhance representations of 'parts') indeed seems to contradict a strictly letter-based (part-based) account. But on the other hand, it has been convincingly demonstrated, both for word and face recognition[12,39], that the identifiability of parts poses a bottleneck on the identification of wholes, and hence, even for the most common words, recognition cannot be truly holistic[12]. Moreover, effects of wholes on the identification of parts are not always facilitatory: facial arrangements, for instance, have been both reported to have positive and negative effects on search performance[40,41]. Developing a theoretical framework that naturally accounts for top–down, contextual enhancement as reported here (see e.g. [23]), while being properly constrained so as to incorporate feature-based bottlenecks, and the occasional detrimental effects of context, provides an important challenge for future research.

A limitation of the current study is that we cannot access sensory representations directly, and instead have to infer them by estimating sensory information from measured neural activity patterns. By itself, the fact that letter identity could be more readily decoded from words than nonwords could in principle merely reflect confounding differences in the BOLD signal between conditions, or in the our ability to extract information from that signal[42]. Importantly, however, we obtained converging evidence using two complementary techniques of probing representational content. One of these (classification) used an independent dataset for training purposes in which only single letters were presented without noise, which suggests that our MVPA techniques were picking up relevant information about the middle letters, rather than irrelevant signals that only covaried with reading (non)words with the respective middle letter. Moreover, enhancement was consistently found in multiple visual areas, was retinotopically specific, but not contingent on exact ROI definitions, and could not be explained by other confounds such as signal amplitude or eye movements. As such, we believe that the most parsimonious explanation of the observed effect is as reflecting an enhancement in the underlying sensory information available to visual cortex itself—in other words, a representational enhancement.

We interpret this representational enhancement as a neural signature of the perceptual enhancement of letters—a process formalised by top–down models of word recognition, and widely characterised in behavioural literature[10,29]. However, a limitation of fMRI is that it cannot differentiate between earlier and later activity. Hence, it is possible that the observed effects arise late, perhaps even much later than what is typically considered 'perceptual' (e.g. > 400 ms); instead perhaps reflecting what one might call iconic memory encoding. Although distinguishing between perceptual and post-perceptual effects is notoriously difficult, future studies might address this by probing perception more directly using an objective measure of perceptual sensitivity, or by using a high temporal resolution method (e.g. ECoG or MEG) in combination with a temporal criterion to arbitrate between perceptual and post-perceptual enhancement of sensory representations.

An apparent disconnect between our study and the existing literature concerns the level of representation at which enhancement occurs. We probed enhancement in early visual cortex (representing visual features such as edges and simple line conjunctions) while in theoretical models[6,14] enhancement is probed at the level of letters, not features. However, perceptual enhancement is a generic mechanism and should not be unique to a specific level of representation. In fact, the main reason[6] that in the classic models enhancement occurs only at the letter level is simplicity: because features comprise the input to the network and are hence not recognised, the possibility of enhancement occurring at the feature level is excluded by design.

Apart from model simplicity, one might argue there are more substantive, cognitive reasons that enhancement is primarily described at the letter level and not at the level of simple features. Specifically, word superiority has been reported with stimuli consisting of mixed case and font[43], implying that sometimes the word superiority effect can be independent of the visual features that define exact letter shape, and may instead act at a more abstract level of letter identity. However, when the exact letter shape is well known and especially under visually noisy conditions (like in our experiment), enhancement of simple low-level visual features appears useful for letter recognition, thereby incentivising top–down enhancement to reach the (functionally well-localised) visual feature level. As such, we do not claim that our experiment shows that word superiority always acts at the level of sensory features. Rather, it demonstrates that in principle these enhancement effects can extend even to the earliest sensory cortical regions, contradicting purely bottom-up accounts in which such top–down enhancement is ruled out by design.

What kind of information is driving the observed representational enhancement effect? A possible source of information is lexical knowledge, although sublexical (orthographic/phonological) knowledge may be an equally plausible candidate. Indeed, behaviourally letters are also more easily recognised when embedded in pronounceable nonwords (pseudowords) than in unpronounceable, orthographically illegal nonwords[44]. Theoretically, such a pseudoword superiority effect can be either understood as originating via top–down connections in a dedicated sublexical route[45] or as arising as a by-product from co-activations of lexical items with overlapping letter combinations[6,46]. The fact that we found increased functional coupling in both VWFA (associated with sublexical orthography) and pMTG (associated with lexical access) makes our data consistent with both types of feedback, originating from the sublexical and the lexical route. Future studies could examine the relative contributions of these forms of feedback, by examining the neural activity patterns to pronounceable pseudowords.

A final point of discussion concerns the interpretation of the information-activation coupling analysis. We interpret the results as pointing to putative candidate sources, because the observed increases in functional coupling match the expected pattern of results if the associated regions were indeed a neural source of the enhancement. However, we acknowledge that since the functional coupling analysis is correlational in nature, the direction of causality implied by this interpretation remains speculative. To get a better understanding of the sources involved this effect, future studies could either directly perturb candidate sources, use a more indirect method for inferring directionality such as laminar fMRI[47] or a directional connectivity analysis[48].

In conclusion, we have observed that word contexts can enhance sensory letter representations in early visual cortex. These results provide the first neural evidence for top–down enhancement of sensory letter representations by word contexts, and suggest that readers can better identify letters in context because they might, quite literally, see them better.

## Methods

**Participants**. Thirty-six participants were recruited from the participant pool at the Donders Centre for Cognitive Neuroimaging. Sample size was chosen to detect a within-subject effect of at least medium size ($d > 0.5$) with 80% power using a two-tailed one-sample or paired $t$-test. The study was in accordance with the institutional guidelines of the local ethical committee (CMO region Arnhem-Nijmegen, The Netherlands, Protocol CMO2014/288), all participants gave informed consent and received monetary compensation. Participants were invited for an fMRI session and a prior behavioural training session, that took place no more than 24 h before the fMRI session. For one participant, who moved excessively between runs, decoding accuracy was never above chance; this participant was excluded from all fMRI analyses. One additional participant had their eyes closed for an extended duration during more than 20 trials, and was excluded from both the behavioural and fMRI

analyses. All remaining participants ($n = 34$, 12 male, mean age $= 23 \pm 3.32$) were included in all analyses. Due to technical problems, one participant only completed four instead of six blocks, all of which were analysed.

**Stimuli**. Stimuli were generated using Psychtoolbox-3 (ref. [49]) running on MATLAB (MathWorks, MA, USA). Stimuli were rear-projected using a calibrated EIKI (EIKI, Rancho Santa Margarita, CA) LC XL 100 projector ($1024 \times 768$, 60 Hz). Each stimulus was a five-letter word or nonword presented in a custom-made monospaced typeface. To prevent that the multivariate analyses would pick up on global low-level features (such as overall luminance or contrast) to discriminate between middle letter identity, the middle characters (U or N) were chosen to be identical in shape and size, but flipped vertically with respect to each other. Words were presented in a large font size, each letter 3.6° wide and with 0.6° spacing between letters. This size was chosen to make the middle letter as large as possible while retaining readability of all letters when fixating at the centre. In addition to the words and nonwords, a fixation dot of 0.8° in diameter was presented at the centre of the screen. To make reading visually challenging and incentivize top–down enhancement of low-level visual features, words were embedded in visual noise. The noise consisted of pixelated squares, each 1.2° wide, offset so that the pixels were misaligned with the letter strokes. Letters were presented on top of the noise with 80% opacity. We chose this type of noise after finding it impacted readability strongly even when the letters were presented at high physical luminance. Brightness values (in the range 0–255) of the noise 'pixels' were randomly sampled from a Gaussian distribution with a mean of 128 and an SD of 50. To make sure that the local brightness was on average identical for each trial and across the screen, the noise patches were generated using a pseudo-random procedure. In each trial, ten noise patches were presented, five of which were independent and randomly generated, while the other five were copies of the random patches, but polarity-inverted in terms of their relative brightness with respect to the mean. This way the brightness of each noise pixel was always 128 (grey) on average in each trial. The order of noise patches was pseudo-random, with the constraint that copied patches were never presented directly before or after their original noise patch. This way the re-use of noise patches was not noticeable and all seemed random.

In the main experiment, we used a blocked design, in which we presented blocks of four long trials (one of each of the four conditions), followed by a null-trial. Each trial was 14-s long, during which ten stimuli were presented. Of those stimuli, nine or occasionally (in 25% of trials) eight were (non-)word items and one or two were (learned) targets. A single presentation consisted of 900 ms of (non) word item plus noise background, and 500 ms of blank screen plus fixation dot (Fig. 1c). Targets were either presented in their regular (learned) form or with one of the non-middle letters permuted, and participants had to discriminate whether the target was regular or permuted. Target correctness and occurrence within the trial were counterbalanced and randomised, with the constraint that targets were never presented directly after each other. The order of word items was shuffled pseudo-randomly, with the constraint that the same letter never repeated twice at the same position (except for the middle letter).

In the functional localiser run, only the middle letters (U and N) plus fixation bulls' eye were presented. We again used a blocked design, with long trials that had a duration of 14 s during which one of letters was repeated at 1 Hz (500 ms on, 500 ms off; see Fig. 1b). During the localiser, each trial was followed by a null-trial in which only the fixation dot was presented for 9.8 s. This was repeated 18 times for each letter.

Two different sets of words and nonwords were used for the training and experimental session. For the experimental session, we used 100 five-letter words with a U or N as third character in Dutch (see Supplementary Table 1), plus equally many nonword items. This particular subset was chosen because they were the 100 most frequent five-letter words with a U or N in Dutch, according to the subtlex database[50]. Each item occurred at least four times and maximally five times (4.2 on average) during the entire experimental session; to ensure repetitions were roughly equally spaced, items were only repeated once all other items were presented equally often. Because we wanted to familiarise participants with the task and the custom-font, but not with the (non)word stimuli themselves (especially because there was considerable variation in the amount of training between participants), we used different (non)words for the training session. For the training session, we used the remaining 50 less frequent five-letter Dutch words with a U and N. For the nonwords, letters were randomly sampled according to the natural frequency of letters in written Dutch[51], with the constraint that adjacent letters were never identical. The resulting nonwords were then hand-selected to ensure all created strings were unpronounceable, orthographically illegal nonwords. The four learned target stimuli were CLUBS and ERNST for the words, and KBUOT and AONKL for the nonwords. These were learned during the prior training session.

**Procedure**. Each participant performed one behavioural training and one experimental fMRI session. The goal of the training was for participants to learn the four target items and learn how to perform the task while maintaining fixation at the centre of the screen. The fMRI session consisted of a brief practice of ~5 min during which the anatomical scan was acquired. This was followed by six experimental runs of 9–10 min, which were followed by a localiser run of ~15 min. We used a blocked design, in which we presented blocks of four long trials (one of each

of the four conditions), followed by a null-trial experimental run consisted of 40 trials of 14 s. Trials were presented in blocks consisting of five trials: one of each condition (U-word, U-nonword, N-word, N-nonword), plus a null-trial during which only the fixation dot was present. The order of trial types within blocks was randomised and equalised: over the entire experiment, each order was presented twice, resulting in a total number of 240 trials (192 excluding nulls). In the functional localiser, single letters were presented blockwise: one letter was presented for 14 s, followed by a null-trial (9.8 s), followed by a trial of the other letter. Which letter came first was randomised and counterbalanced across participants.

**Statistical testing**. For each (paired/one-sample) statistical comparison, we first verified that the distribution of the data did not violate normality and was outlier free, determined by the D'Agostino and Pearson's test implemented in SciPy and the 1.5 IQR criterion, respectively. If both criteria were met, we used a parametric test (e.g. paired $t$-test); otherwise, we resorted to a non-parametric alternative (e.g. Wilcoxon sign rank). All statistical tests were two-tailed and used an alpha of 0.05. For effect sizes, we report Cohen's $d$ for the parametric and biserial correlations for the non-parametric tests.

**fMRI acquisition**. Functional and anatomical images were collected with a 3T Skyra MRI system (Siemens), using a 32-channel headcoil. Functional images were acquired using a whole-brain T2*-weighted multiband-4 sequence (TR/TE = 1400/33.03 ms, voxel size = 2 mm isotropic, 75° flip angle, A/P phase encoding direction). Anatomical images were acquired with a T1-weighted MP-RAGE (GRAPPA acceleration factor = 2, TR/TE = 2300/3.03 ms, voxel size 1 mm isotropic, 8° flip angle).

**fMRI preprocessing**. fMRI data pre-processing was performed using FSL 5.0.11 (FMRIB Software Library; Oxford, UK[52]). The pre-processing pipeline included brain extraction (BET), motion correction (MCFLIRT), temporal high-pass filtering (128 s). For the univariate and univariate-multivariate coupling analyses, data were spatially smoothed with a Gaussian kernel (4 mm FWHM). For the multivariate analysis, no spatial smoothing was applied. Functional images were registered to the anatomical image using boundary-based registration as implemented in FLIRT and subsequently to the MNI152 T1 2-mm template brain using linear registration with 12 degrees of freedom. For each run, the first four volumes were discarded to allow for signal stabilisation. Most FSL routines were accessed using the nipype framework[53]. Using simple linear registration to align between participants can result in decreased sensitivity compared to more sophisticated methods like cortex-based alignment[54]. However, note that using a different inter-subject alignment method would not affect any of the main analyses, which were all performed in native EPI space. The only analysis that could be affected is the whole-brain version of the information-activation coupling analysis (Fig. 4c; Supplementary Fig. 11). However, this was only an exploratory follow-up on the pre-defined ROI-based coupling analysis, intended to identify potential other regions displaying the signature increase in coupling. For this purpose, the simple linear method was deemed appropriate.

**Univariate data analysis**. To test for differences in univariate signal amplitude between conditions, voxelwise GLMs were fit to each run's data using FSL FEAT. For the experimental runs, GLMs included four regressors of interest, one for each condition (U-word, U-nonword, etc). For the functional localiser runs, GLMs included two regressors of interest (U, N). Regressors of interest were modelled as binary factors and convolved with a double-gamma HRF. In addition, (nuisance) regressors were added for the first-order temporal derivatives of the regressors of interest, and 24 motion regressors (six motion parameters plus their Volterra expansion, following Friston et al.[55]). Data were combined across runs using FSL's fixed-effects analysis. All reported univariate analyses were performed on an ROI basis by averaging all parameter estimates within a region of interest, and then comparing conditions within participants (see Supplementary Figs. 4, 5).

**Multivariate data analysis**. For the multivariate analyses, spatially non-smoothed, motion-corrected, high-pass filtered (128 s) data were obtained for each ROI (see below for ROI definitions). Data were temporally filtered using a third-order Savitzky-Golay low-pass filter (window length 21) and z-scored for each run separately. Resulting timecourses were shifted by three TRs (i.e. 4.2 s) to compensate for HRF lag, averaged over trials, and null-trials discarded. For each participant, this resulted in 18 samples per class for the localiser (i.e. training data) and 96 samples per condition (word/nonword) for the main runs (i.e. testing data).

For the classification analysis, we used a logistic regression classifier implemented in sklearn 0.2 (ref. [56]) with all default settings. The model was trained on the time-averaged data from the functional localiser run and tested on the time-averaged data from the experimental runs. Because we had the same number of samples for each class, binary classification performance was evaluated using accuracy (%).

For the pattern correlation analysis, only the time-averaged data from the main experiment were used. Data were randomly grouped into two arbitrary splits that both contained an equal number of trials of all four conditions (U-word, U-nonword, N-word, N-nonword). Within each split, the time-averaged data of each trial were again averaged to obtain a single average response for each

condition per split. For both word/nonword conditions separately, these average responses were then correlated across splits. This resulted, for both word and nonword conditions, in two (Pearson) correlation coefficients: $\rho_{within}$ and $\rho_{between}$, obtained by correlating the average response to stimuli with the same or different middle letter, respectively. This process was repeated 12 times, each time using a different random split of the data, and all correlation coefficients were averaged to obtain a single coefficient per comparison, per condition, per participant. Finally, pattern letter information for each condition was quantified by subtracting the two average correlation coefficients ($\rho_{within} - \rho_{between}$).

For the searchlight variant of the multivariate analyses, we performed exactly the same procedure as described in the manuscript. However, instead of using a limited number of a priori defined ROIs, we used a spherical searchlight ROI that slid across the brain. A searchlight radius of 6 mm was used, yielding an ROI size of about 170 voxels on average, similar to the 200 voxels in our main ROI. For both analyses, this resulted in a map for each outcome metric for each condition for each subject, defined in native EPI space. These maps were then used for subsequent analyses (see Supplementary Note 1).

**Information-activation coupling analysis**. For the information-activation coupling analysis, we used a GLM-based approach to predict regional BOLD amplitude as a function of early visual cortex classification evidence, and tested for an increase in coupling (slope) for words compared to nonwords (see Fig. 4b). The GLM had one variable of interest, visual cortex classification evidence (see below for definition) that was defined on a TR-by-TR basis, and split over two regressors, corresponding to both conditions (word/nonword). In addition, first-order temporal derivatives of the two regressors of interest and the full set of motion regressors (from the FSL FEAT GLM) were included to capture variability in HRF response onset and motion-related nuisance signals, respectively. Because the classification evidence was undefined for null-trials, these were omitted. To compensate for temporal autocorrelation in the data, pre-whitening of the data was applied using the AR(1) noise model as implemented in nistats[56]. The resulting GLM yielded two regression coefficients (one per condition) for each participant, which were then compared at the group level to test for an increase in coupling in word contexts. Conceptually, this way of testing for condition-dependent changes in functional coupling is analogous to PPI[16] but using a multivariate time-course as a 'seed'. This timecourse, classification evidence, was defined as the probability assigned by the logistic regression model to the correct outcome—or $\hat{p}(A|y = A)$. This probabilistic definition combines aspects of both prediction accuracy and confidence into a single quantity. Mathematically it is defined, as in any binomial logistic regression classifier, via the logistic sigmoidal function:

$$\hat{p}(A|y = A) = \begin{cases} \frac{1}{1 + e^{-\theta^T X}} & \text{if } y = 1 \\ 1 - \frac{1}{1 + e^{-\theta^T X}} & \text{if } y = 0 \end{cases}, \tag{1}$$

where $\theta$ are the model weights, $y$ is the binary stimulus category, $\mathbf{X}$ are the voxel response patterns for all trials, and the letter 'U' is coded as 1 and 'N' as 0. Note that while the value of $\hat{p}(A|y = A)$ itself is bounded between 0 and 1, the respective regressors were not after applying prewhitening to the design matrix (see Fig. 4b).

Two variants of the GLM analysis were performed: one on timecourses extracted from two candidate ROIs and one on each voxel independently. For the ROI-based approach, timecourses were extracted by taking the average timecourse of all amplitude-normalised (z-scored) data from two ROIs: left pMTG and VWFA (see 'ROI definition' for details). For the brain-wide variant, the same GLM was estimated voxelwise for each voxel independently. This resulted in a map with the difference in coupling parameters for each voxel, for each participant ($\beta_{word} - \beta_{nonword}$) in native MRI space. These maps were then transformed to MNI space, after which a right-tailed one-sample $t$-test was preformed to test for voxels showing an increase in coupling in word conditions. The resulting $p$-map was converted into a $z$-map and thresholded using FSL's Gaussian random-field-based cluster thresholding, using the default cluster-forming threshold of $z > 3.1$ (i.e., $p < 0.001$) and a cluster significance threshold of $p < 0.05$.

**ROI definition**. For the ROIs of V1–V4, fusiform cortex and inferior temporal cortex, Freesurfer 6.0 (ref. [57]) was used to extract labels (left and right) per subject based on their anatomical image, which were transformed to native space and combined into a bilateral mask. Labels for V1–V2 were obtained from the default atlas[58], whereas V3 and V4 were obtained from Freesurfer's visuotopic atlas[59]. Early visual cortex (EVC) was defined as the union of V1 and V2.

The VWFA was functionally defined following a procedure based on earlier work[34]. Briefly, first we took the union of left fusiform cortex and left inferior temporal cortex that were defined via individual cortical parcellations obtained from freesurfer, and trimmed the anterior parts of the resulting mask. Within this broad, left-lateralised ROI, we then selected the 200 voxels that were most selective to words over nonwords (i.e. words over orthographically illegal, unpronounceable letter strings) as defined by the highest Z-statistics in the respective word–nonword contrast in the univariate GLM. Similarly to Kay and Yeatman[34], we found that for most participants this resulted in a single, contiguous mask and in other participants in multiple word-selective patches. There are two main reasons we used the simple contrast word–nonword from the main experiment, rather than running a separate, dedicated VWFA localiser. First, using the main task strongly

increased statistical power per subject as we could use a full hour of data per participant to localise VWFA. Second, the comparison of words and unpronounceable letter strings (with matched unigram letter frequency) solely targets regions that are selective to lexical and orthographic information (i.e. the more anterior parts of VWFA, according to the VWFA hierarchy reported in ref. [32]). As such, the localiser only targets regions selective to the type of linguistic (lexical or orthographic) knowledge that could underlie the observed effect. This stands in contrast to other, less-restrictive VWFA definitions (such as words > phase scrambled words, or words > false fonts).

For the multivariate stimulus representation analyses, we did not use the entire anatomical ROIs defined above, but performed a selectivity-selection to ensure we probed voxels that were selective to the relevant part of the visual field. In this procedure, we defined the most selective voxels as those with the $k$ highest $Z$-statistics when we contrasted any letter (U or N) versus baseline in the functional localiser GLM. Following ref. [15], we took 200 voxels as our predefined value for $k$. To verify that our results were not contingent on this specific (but arbitrary) value, we also made a large range of masks for early visual cortex by varying $k$ between 50 and 1000 with steps of 10. Repeating the classification and pattern correlation analyses over all these masks revealed that the same pattern of effects was obtained over almost the full range of mask definitions, and that the best classification performance was in fact at our predefined value of $k = 200$ (Supplementary Fig. 3).

For the peripheral visual ROI, voxels were selected based on the functional criterion that they showed a strong response to stimuli in the main experiment (which spanned a large part of the visual field), but a weak or no response to stimuli in the localiser (which were presented near fixation). Specifically, voxels were selected if they were both in the top 50% of $Z$-stats for the contrast visual stimulation > baseline in the main experiment, and in the bottom 50% of $Z$-scores for visuals stimulation > baseline in the localiser. This resulted in masks that contained on average 183 voxels, similar to the 200 voxels in the central ROI. In our initial analysis, we focussed on V1 (see Supplementary Fig. 9) because it has the strongest retinotopy. However, the same was also applied to early visual cortex with similar results (see Supplementary Note 1).

To define pMTG, we performed an automated meta-analysis using Neurosynth[60]. Because we were interested in pMTG as a hub for lexical access, we searched for the keyword 'semantic'. This resulted in a contrast map based on 1031 studies which we thresholded at an arbitrarily high $Z$-value of $Z > 9$. The resulting map was mainly restricted to two hubs, in the IFG and pMTG. We selected left pMTG by overlaying the map with an anatomical mask of medial temporal gyrus from FSL's Harvard-Oxford Atlas. The resulting map was brought to native space by applying the registration matrix for each participant.

**Behavioural data analysis.** Participants had 1.5 s after target onset to respond. Reaction times under 100 ms were considered spurious and discarded. If two non-spurious responses were given, only the first response was considered and evaluated. Median reaction times and mean accuracies were computed for both (word and nonword) conditions and compared within participants.

**Eye tracking.** Eye movements were recorded using an SMI iView X eye monitor with a sampling rate of 50 Hz. Data were pre-processed and submitted to two analyses: number of trials during which eyes were closed for extended periods, and comparison of horizontal (reading-related) eye movements between conditions.

During pre-processing, all data points during which there was no signal (i.e. values were 0) were omitted. After omitting periods with no signal, data points with spurious, extreme values (which sometimes occurred just before or after signal loss) were omitted. To determine which values were spurious or extreme, we computed the z-score for each points, over the entire run and ignoring the periods where signal was 0, and considered all values higher than 4 extreme and spurious. Similar to the periods with no signal, these timepoints were also omitted in following analysis. The resulting 'cleaned' timecourses were then visually inspected to evaluate their quality. For two participants, the data were of insufficient quality to include in any analysis. For six participants, there were enough data of sufficient quality to perform the overall amount of reading-related eye movements between conditions, but signal quality was insufficient to quantify the number of trials during which the eyes were shut for an extended period. This is because in these participants there were various periods of intermittent signal loss that were related to signal quality, not to the eyes being closed. To compare eye movements between conditions, we took the standard deviance of the gaze position over the reading (horizontal) direction, and averaged this over each trial. Because the resulting data contained outliers (i.e. trials during which the participants failed to maintain fixation), we took the median over trials in each condition (word/nonword), and compared them within participants (Supplementary Fig. 6). For the participants where the data were consistently of sufficient quality, periods of signal loss longer than 1.2 s were considered 'eyes closed for extended period'. As an inclusion criterion, we allowed no more than 25 trials during which eyes were closed for an extended period. This led to the exclusion of one participant, who had 33 trials during which the eyes were closed for an extended period. This participant was a clear outlier: of all participants with sufficient quality eye tracking data to be included in this analysis, 14 had no trials during which eyes were closed for an extended period, and in the remaining 12 with at least one such trial the median number of trials was 3.5.

**Neural network model.** Simulations were performed using a predictive coding formulation of the classic interactive activation model[6,7]. We begin by explaining the model at an abstract level, then outline the algorithmic and mathematical details in generic terms, and then specify the exact settings we used for our model architecture, and how we used them in our simulations.

The interactive activation model is a hierarchical neural network model which takes visual features as inputs, integrates these features to recognise letters, and then integrates letters to recognise words. Critically, activity in word-units is propagated back to the letter-level, making the letter detectors sensitive not only to the presence of features (such as the vertical bar in the letter E), but also to neighbouring letters (such as the orthographic context HOUS_ preceding the letter E). This provides a top–down explanation for context effects in letter perception, such as (pseudo)word superiority. The predictive coding formulation of this model was first described by Spratling[14]. It uses a particular implementation of predictive coding—the PC/BC-DIM algorithm—that reformulates predictive coding (PC) to make it compatible with Biased Competition (BC) and uses Divisive Input Modulation (DIM) as the method for updating error and prediction activations. The goal of the network is to infer the hidden cause of a given pattern of inputs (e.g. the 'hidden' letter underlying a pattern of visual features) and create an internal reconstruction of the input. Note that the reconstruction is model-driven and not a copy of the input. Indeed, when the input is noisy or incomplete, the reconstruction will ideally be a denoised or pattern-completed version of the input pattern. Inference can be done hierarchically: at the letter-level, predictions represent latent letters given patterns of features, whilst at the word-level predictions represent latent words given patterns of letters (and reconstructions, inversely, represent reconstructed patterns of letters given the predicted word).

Mathematically, the network can be conveniently described as consisting of three components: prediction units ($\mathbf{y}$), reconstruction units ($\mathbf{r}$) and error units ($\mathbf{e}$) that can be captured in only three equations. First, at each level, error units combine the input pattern ($\mathbf{x}$) and the reconstruction of the input ($\mathbf{r}$) to compute the prediction error ($\mathbf{e}$):

$$\mathbf{e} = \mathbf{x} \oslash [\mathbf{r}]_{\epsilon_2}. \tag{2}$$

Here, $\mathbf{x}$ is a ($m$ by 1) input vector; $\mathbf{r}$ is a ($m$ by 1) vector of reconstructed input activations, $\oslash$ denotes pointwise division and the square brackets denote a max operator: $[v]_\epsilon = \max(\epsilon, v)$. This max-operator prevents division-by-zero errors when all prediction units are silent and there is no reconstruction. Following Spratling[14], we set $\epsilon_2$ at $1 \times 10^{-3}$. Division sets the algorithm apart from other versions of predictive coding that use subtraction to calculate the error (see Spratling[61] for review). The prediction is computed from the error via pointwise and matrix multiplication:

$$\mathbf{y} \leftarrow [\mathbf{y}]_{\epsilon_1} \otimes W\mathbf{e}. \tag{3}$$

Here, $W$ is a ($n$ by $m$) matrix of feedforward weights that map inputs onto latent causes (e.g. letters), $\otimes$ denotes pointwise multiplication, square brackets represents a max operator and $\epsilon_1$ is set at $1 \times 10^{-6}$. Each row of $W$ maps the pattern of inputs to a specific prediction unit representing a specific latent cause (such as the letter) and can hence be thought of as the 'preferred stimulus' or basis vector for that prediction unit. The entire $W$ matrix is then best thought of as comprising the layer's model of its environment. Finally, from the distribution of activities of the prediction units ($\mathbf{y}$), the reconstruction of expected input features ($\mathbf{r}$) is calculated as a simple linear generative model:

$$\mathbf{r} = V\mathbf{y}, \tag{4}$$

where $V$ is a ($m$ by $n$) matrix of feedback weights that map predicted latent causes (e.g. letters) back to their elementary features (e.g. strokes) to create an internal reconstruction of the predicted input, given the current state estimate. As in many multilayer networks, the model adheres to a form of weight symmetry: $V$ is almost identical to $W^T$, but its values are values normalised so that each column sums to one. To perform inference, prediction units can be initialised at zero (or with random values) and the Eqs. (2–4) are updated iteratively. To perform top–down hierarchical inference, reconstructions from a higher-order stage (e.g. recognising words) can be sent back to the lower-order stage (e.g. recognising letters) as additional input. To accommodate these recurrent inputs, additional weights have to be defined that are added to $W$ and $V$ as extra columns and rows, respectively. The strength of these weights is scaled to control the reliance on top–down predictions.

**Architecture specification.** The interactive activation architecture we used was a modification of the network described and implemented by Spratling[14], extended to recognise five-letter words, trained on the Dutch subtlex vocabulary, and with a slight change in letter composition. Letters are presented to the network using a simulated font adapted from the one described by Rumelhart and Siple[62] that composes any character using 14 strokes (Supplementary Fig. 12). For our five-letter network, the input layer comprises five 14-dimensional vectors (one per character) that each represent the presence of 14 line segments for one letter position. Note that conceptually it is easier to partition the input into five 14-dimensional vectors, in reality these were concatenated into a single 70-dimensional vector $\mathbf{x}$.

At the first level, weight matrix **W** has 180 rows 250 columns: rows comprise five slots of 36 alphanumeric units ($5 \times 36 = 180$); the first columns comprise five slots of 14 input features ($5 \times 14 = 70$) and the last 180 columns route the top–down reconstruction from the word level. To define the weights of 70 (feedforward) columns, we used encoding function $\phi(c)$ that takes an alphanumeric character and maps it into a binary visual feature vector. For each alphanumeric character, the resulting feature vector was concatenated five times and the resulting 70 dimensional vector comprised the first row. This was repeated for all 36 alphanumeric characters and concatenated five times. The resulting numbers were then normalised so that the columns summed to one. Then we added the weights of the second 180 columns (inter-regional feedback coming from $5 \times 36$ letter reconstructions) were simply a 180 by 180 identity matrix multiplied by a scaling factor to control top–down strength. For our 'top–down model' (Fig. 3b), we set the scaling factor at 0.4; in the 'bottom-up model', we set it to $10^{-6}$ to effectively cancel the influence of feedback, resulting in a 'bottom-up' model. At the second level, weight matrix **W** had 6778 rows and 180 columns, representing 6776 Dutch five-letter words from the subtlex corpus, plus the two learned nonword targets (that we included in the vocabulary as participants learned these during training) and five times 36 alphanumeric characters. The orthographic frequency of letters as specified by the corpus was hard coded into the weights and then normalised to sum to one.

Although there are substantial implementational differences between this model and the classic connectionist version of the interactive activation model[6,7], the version described here has been shown to capture all key experimental phenomena of the original model (see ref. 14 for details). Since our simulations only tried to validate and demonstrate a qualitative principle, not subtle quantitative effects, the exact numerical differences related to the differences in implementation should not matter for the effect we demonstrate here.

**Simulations**. Because our paradigm is different from classical paradigms, we performed simulations to confirm that the top–down account indeed predicts the representational enhancement we set out to detect. Although the main simulation result (Fig. 3a) is not novel, our simulation, by mirroring our paradigm, departs from earlier simulations in some aspects, which we will clarify before going into the implementation details. First, most word superiority studies present stimuli near-threshold: words are presented briefly, followed by a mask, and average identification accuracies typically lie between 60 and 80%. This is mirrored in most classic simulations, where stimuli are presented to the network for a limited number of iterations and followed by a mask, leading to similar predicted response accuracies[7,14]. In our task, stimuli are presented for almost a second, and at least the critical middle letter is always clearly visible. This is mirrored in our simulations, where stimuli are presented to the network until convergence and predicted response accuracies of the network are virtually 100% in all conditions (see Supplementary Fig. 2). As such, an important aspect to verify was that enhancement of a critical letter can still occur when it is well-above threshold and response accuracy would be virtually at 100% already. Second, our simulations used the same Dutch word and nonword materials used in the experiment. This includes the occurrence of learned targets in the nonword condition, which we added to the vocabulary of the network and were hence a source of contamination as 12% of the items in the nonword condition were in fact in the vocabulary. Finally, unlike classical simulations, stimuli were corrupted by visual noise.

For Fig. 3a, we simulated 34 artificial 'runs'. In each run, 48 words and 48 nonwords were presented to a network with feedback connections (feedback weight strength 0.4) and without word-to-letter feedback (feedback weight strength $10^{-6}$). The same Dutch, five-letter (non)words were used as in the main experiment, and like in the experiment 12% of the (non)word items were replaced by target items. Critically, the nonword targets were learned and hence were part of the vocabulary of the network. To present a (non)word to the network, each character $c$ has to be first encoded into a set of visual features and then corrupted by visual noise to produce an input vector **x**:

$$\mathbf{x} = \phi(c) + \mathcal{N}(\mu, \sigma^2). \tag{5}$$

For $\mu$ we used 0, $\sigma$ was set to 0.125, and any values of **x** that became negative after adding white noise were zeroed. The network then tried to recognise the word by iteratively updating its activations using Eqs. (2–4), for 60 iterations. To compute the 'relative evidence' metric we used in Fig. 3a to quantify representational quality $q(\mathbf{y})$, we simply take the fraction of activation for the correct letter ($\mathbf{y}_i$) of the sum of letter activations for all characters at the third slot:

$$q(\mathbf{y}) = \frac{\mathbf{y}_i}{\sum_{j=37}^{73} \mathbf{y}_j}. \tag{6}$$

Finally, to compute predicted response probabilities as in Supplementary Fig. 2, we followed McClelland and Rumelhart to use Luce's rule to compute responses probabilistically:

$$p(R_i) = \frac{e^{\beta \mathbf{y}_i}}{\sum_{j=37}^{73} e^{\beta \mathbf{y}_j}}. \tag{7}$$

The $\beta$ parameter (or inverse softmax temperature) determines how rapidly the response probability grows as $\mathbf{y}_i$ increases (i.e. the 'hardness' of the argmax operation) and was set at 10, following McClelland and Rumelhart[6,7], but results

are similar for any typical beta value that is approximately in the same order of magnitude.

All simulations were performed using custom MATLAB code, which was an adaptation and extension of the MATLAB implementation published by Spratling[14].

**Reporting summary**. Further information on research design is available in the Nature Research Reporting Summary linked to this article.

## Data availability
All raw data required to reproduce all analyses and figures are uploaded onto the Donders Data Repository and can be found at http://hdl.handle.net/11633/aacjymw7. A reporting summary for this Article is available as a Supplementary Information file.

## Code availability
All code required to reproduce all analyses and figures are uploaded onto the Donders Data Repository and can be found at http://hdl.handle.net/11633/aacjymw7.

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

## Acknowledgements

This work was supported by The Netherlands Organisation for Scientific Research (NWO Research Talent grant to M.H.; NWO Vidi grant to F.P.d.L.; 016.Veni.195.435 to M.E.; Gravitation Program Grant Language in Interaction no. 024.001.006 to P.H.) and the European Union Horizon 2020 Program (ERC Starting Grant 678286, "Contextvision" to F.P.d.L). We thank Ashley Lewis for helpful comments on and discussions of an earlier version of this manuscript.

## Author contributions

M.H., F.P.d.L., P.H., D.R. and M.E. designed the study. M.H. and D.R. collected the data. M.H., D.R., M.E. and F.P.d.L. conceived of the analysis plan. M.H. analysed the data. M.H. performed simulations. M.H. wrote the initial draft. All authors contributed to the final manuscript.

## Competing interests

The authors declare no competing interests.
