## [Peer Review File · Nature Communications]

Reviewers' Comments:

Reviewer #1:

Remarks to the Author:

This article presents an empirical investigation into the cortical mechanisms underlying the word-superiority effect. Specifically, the results show top-down facilitation of neural representations corresponding to a specific letter in the early visual cortex when that letter appears in a word compared to when that letter appears within a non-word. These results are interesting and important. Furthermore, the manuscript is clearly written and the work seems to be methodologically sound. I am therefore happy to recommend publication, but have a few suggestions for minor improvements if a revision is necessary:

1) The current results are concerned with the word-superiority effect. However, this can be seen as a specific example of a more general phenomenon: contextual influences on visual processing. Such contextual effects have been explored extensively, for example, much work has been published on the effects of surrounds and extended contours on the response properties of orientation selective neurons in V1. Similarly, the effects of distant, contextual, parts of an image have been recorded for border-ownership cells in V2. If space allows, it would be interesting if the authors could relate the current results to this previous work, and speculate about whether or not they are likely to involve similar mechanisms. Some relevant references might include:

Kveraga et al, *Brain and Cognition* 65, 2007

Przybylski, *Current Biology* 8, 1998

Phillips et al, *Neuroscience and Biobehavioral Reviews*, 52, 2015

Khan and Hofer, *Current Opinion in Neurobiology*, 52, 2018

Muckli et al, *Current Biology*, 25, 2015

Smith and Muckli, *PNAS*, 107, 2010

Gilbert et al, *Vision Research*, 40, 2000

Zipser et al, *Journal of Neuroscience*, 16, 1996

Lamme and Roelfsema, *TINS*, 23, 2000

Angelucci and Bressloff, *Progress in Brain Research*, 154, 2006

2) While the methods section comprehensively, and clearly, describes the experimental procedures that have been used, there are a couple of places in the main text where these procedures have been summarised in such a way that there is some ambiguity. Particularly, from the text starting at line 19 on page 3 it is unclear if a single block of stimuli can contain both words and non-words, or if words and non-words only appear in separate blocks. Also, the reference given at line 18 on page 4 suggests that model shown in Fig 2a is that of McClelland and Rumelhart rather than a very different model (which produces similar behaviour) proposed by Spratling.

3) At the end of the first paragraph of the Discussion, the authors claim that the results are consistent with predictive coding. This claim could do with some additional explanation, especially, as predictive coding proposes that top-down influences are inhibitory. Hence, in a predictive coding model one would expect that identification of a word at one level of the hierarchy would result in suppression of responses to the letters making up that word in earlier levels of the hierarchy. This seems to be difficult to reconcile with the current results.

4) There are some inconsistencies in the use of bold type in the text and equations on pages 25 and 26: variables y and r , are sometimes written in bold, and sometimes not.

Reviewer #2:

Remarks to the Author:

The present work uses an fMRI decoding approach to understand the long debated neural mechanisms underlying the word superiority effect. The experimental design is elegant and the results provide an interesting account of changes in the neural representation of individual letters that are embedded in words as compared to pseudowords. Overall I found the results compelling and the methods sound. There are two major points that I would like to see addressed in a revision and four additional points that I think merit additional consideration.

Sincerely,

Jason Yeatman

First, the VWFA, in fact, responds more to pronounceable pseudowords than real words. Conventionally the VWFA is localized by comparing the response to words against the response to other image categories. So by defining cortex that responds more to words than pronounceable pseudowords it is unclear what region is actually being localized (it has been reported that the anterior fusiform responds more to words than pseudowords but this is still not a conventional localizer). This needs further investigation. Ideally they would use another localizer as critical components of their interpretation rest on this region of the reading network. However, if that is not possible, then more needs to be done to show what is being localized in individual subjects. Are they finding a consistent region in each individual? How does the region they are finding relate to the broader literature? Where is this region positioned anatomically relative to the typical location of the VWFA (the VWFA has a very stereotyped location in the occipitotemporal sulcus)? A general point is that the manuscript would benefit from integrating a bit more of the literature on the VWFA.

Second, the theory presented in the introduction rests on the assertion that representations are changing in early visual cortex as opposed to, for example, a change in SNR across much of the brain. Can the authors conduct a searchlight analysis showing correlation and classification accuracy across the cortex (ideally using a similar size searchlight to early visual cortex ROI)? Does this show a peak in early visual cortex (specifically around fovea)? Or can U and N be decoded throughout much of the brain and is the representation different for words versus pseudowords? If so, how does this affect the theory?

Other suggestions/questions:

It would be helpful for me to see the bold amplitudes in the main text rather than just the supplement and to show the mean difference between conditions (in addition to the violin plots). The violin plots make it clear that there are large inter-subject differences in BOLD amplitude but it is hard for me to see how consistent the mean difference between conditions is. This is a very minor point.

The whole-brain results shown in figure 4c appear to use a drawing of the brain. Could the authors switch to a more conventional approach such as a cortical surface derived from freesurfer?

Rather than using an affine alignment for the group results, cortex based alignment would be preferable. Particularly in ventral temporal cortex, inter-subject variability is accounted for much better by cortex based alignment than volume based alignment (and a simple affine will leave a lot of anatomical variability). Work by Frost and Goebel as well as work from Weiner and Grill-Spector has shown the large benefits of cortex based alignment within ventral temporal cortex.

In the main experiment the words are embedded in noise. What is the significance of this decision and

how much do the results depend on the stimulus being embedded in noise?

Small notes:

p3: what is a target? is a spelling task an orthographic task?

p3: behavioral result: what are the effect sizes, in units of RT (ms) and accuracy?

p5, line 9: what does it mean that the SNR was identical? SNR of what?

To what extent is the ROI stimulated by letters to the side and does that matter?

Reviewer #3:

Remarks to the Author:

The authors report an investigation into the neural bases of contextual effects in perception. Specifically, they test an account of the word-superiority effect (better representation of individual letters in context of a real world vs non-word) that is based on top-down influences on perception. This is distinguished from a post-perceptual, decisional account of the phenomena. Participants viewed words and nonwords, each with one of two possible central letters. fMRI classification measures showed that discrimination of the central letter was improved, for a range of early visual-cortex ROIs, in words vs non-words. Further, a PPI-style analysis identified candidate neural sources of hypothesised top-down effects in visual word form area, PMTG, and IFG – all regions with good a priori reasons to be implicated in word processing. Control analyses excluded influences of main-effect (univariate) differences on the classification results, and differences in eye movements. The authors conclude that contextual effects on letter perception must at least in part be due to top-down facilitation of early perceptual processes.

The manuscript is clear and thorough, and addresses a problem of both long-standing and current interest. Numerous strengths are in evidence, for example: a relatively large N compared to typical studies; the use of independent localisers and hypothesis-driven regions of interest (and the latter tested over a range of voxel numbers); the use of a formal modelling approach to establish connection between theory and data; careful balancing of the two stimulus classes; and testing eye movements to rule out potential confounds in attention to words vs nonwords.

The main limitation in my view is that the authors' logic implicitly assumes that evidence of representational quality effects in retinotopic visual cortex are evidence only for a top-down effect on "purely" perceptual representations. This logic should be further exposed – that is, the authors should elaborate on the argument that other non-perceptual (e.g. decision-related) effects could not in principle manifest as influences on early visual cortical areas. Here we must keep in mind that obviously BOLD activity captures the full time-course of activity elicited by the word/non-word stimuli and is not differentially sensitive to earlier vs later waves of neural activity.

Second, the authors note that "enhancement was consistently found in multiple visual areas". As a complement, to further demonstrate the specificity of the context effect, it would have been useful to identify a control visual region(s) where differential effects of context would not be expected. For example, given the strong retinotopy of V1, it should be possible to distinguish foveal voxels that are presumably the main target of the central U and N stimuli, from more peri-foveal or peripheral voxels, that presumably would not be so strongly influenced by the interaction of the central item with the word/non-word context. A demonstration of such retinotopic specificity would strengthen the authors' claims that what is being influenced by context is indeed the specific, early perceptual representation of the critical items.

Specific points:

It is not clear from the Methods whether the U/N localiser took place before or after the main experiment. If it took place first, this could have had the effect of biasing participants' attention towards the central U/N in the main task. While that in itself would not explain the current findings, it could have the effect of unnaturally increasing the strength of that central letter's neural representation in a way that would not be the case in natural word reading.

Page 13 A further complication to this picture, not discussed by the authors, is found in evidence for "contextual suppression" e.g. Suzuki and Cavanagh, 1995 found that whole-face configuration can impede perceptual access to individual features. Consistent context is not always facilitatory – and so in some cases early visual representations of individual elements should be weaker within a meaningful context, rather than stronger.

We are pleased with the positive assessment of our manuscript and would like to thank all reviewers for their constructive comments. We have addressed the reviewers' comments in a detailed, point-by-point reply below.

Reviewer #1

This article presents an empirical investigation into the cortical mechanisms underlying the word-superiority effect. Specifically, the results show top-down facilitation of neural representations corresponding to a specific letter in the early visual cortex when that letter appears in a word compared to when that letter appears within a non-word. These results are interesting and important. Furthermore, the manuscript is clearly written and the work seems to be methodologically sound. I am therefore happy to recommend publication, but have a few suggestions for minor improvements if a revision is necessary:

We want to thank the reviewer for their positive assessment of our work.

1) The current results are concerned with the word-superiority effect. However, this can be seen as a specific example of a more general phenomenon: contextual influences on visual processing. Such contextual effects have been explored extensively, for example, much work has been published on the effects of surrounds and extended contours on the response properties of orientation selective neurons in V1. Similarly, the effects of distant, contextual, parts of an image have been recorded for border-ownership cells in V2. If space allows, it would be interesting if the authors could relate the current results to this previous work, and speculate about whether or not they are likely to involve similar mechanisms. Some relevant references might include:

Kveraga et al, Brain and Cognition 65, 2007

Przybylski, Current Biology 8, 1998

Phillips et al, Neuroscience and Biobehavioral Reviews, 52, 2015

Khan and Hofer, Current Opinion in Neurobiology, 52, 2018

Muckli et al, Current Biology, 25, 2015

Smith and Muckli, PNAS, 107, 2010

Gilbert et al, Vision Research, 40, 2000

Zipser et al, Journal of Neuroscience, 16, 1996

Lamme and Roelfsema, TINS, 23, 2000

Angelucci and Bressloff, Progress in Brain Research, 154, 2006

We agree with the reviewer and have now extended our discussion on the wider literature on contextual influences on perception (p13).

2) While the methods section comprehensively, and clearly, describes the experimental procedures that have been used, there are a couple of places in the main text where these procedures have been summarised in such a way that there is some ambiguity. Particularly, from the text starting at line 19 on page 3 it is unclear if a single block of stimuli can contain both words and non-words, or if words and non-words only appear in separate blocks. Also, the reference given at line 18 on page 4 suggests that model shown in Fig 2a is that of McClelland and Rumelhart rather than a very different model (which produces similar behaviour) proposed by Spratling.

We want to thank the reviewer for pointing us to these ambiguities. We have now clarified both issues in the main text (see P3 and P4).

3) At the end of the first paragraph of the Discussion, the authors claim that the results are consistent with predictive coding. This claim could do with some additional explanation, especially, as predictive coding proposes that top-down influences are inhibitory. Hence, in a predictive coding model one would expect that identification of a word at one level of the hierarchy would result in suppression of responses to the letters making up that word in earlier levels of the hierarchy. This seems to be difficult to reconcile with the current results.

The reviewer raises an important issue, which we may not have unpacked sufficiently in the previous version of the manuscript. According to predictive coding theories, each cortical region contains two sub-populations of neurons: 1) prediction error neurons and 2) prediction/representation neurons. The reviewer is correct that in predictive coding theories, top-down connections have a (polysynaptically) inhibitory effect. However, this inhibition acts

only on the representation neurons that are inconsistent with the higher-order interpretation and, as a result, on the prediction error neurons (assuming that the top-down interpretation is correct and decreases the error). As such, a representational enhancement effect is in fact well in line with predictive coding theory – at least for the prediction units. This is also what we probe in the simulation, when we quantify representational strength of the prediction neurons.

The fact that predictive coding theory can make different empirical predictions for different neuron types clearly introduces some empirical flexibility. Moreover, we did not observe a univariate suppression effect for words compared to nonwords, which would also be an important hallmark for predictive coding models. For these reasons we do not want to commit ourselves to a predictive coding interpretation, but rather see predictive coding as merely one of the top-down perceptual inference frameworks that is in line with our results.

For a recent discussion of how representational enhancement fits into predictive coding models (in this case the more conventional predictive coding models), see Friston (2018). For a discussion in connection with an empirical observation of an enhancement effect in combination with a univariate suppression that is a signature of predictive coding, see Kok et al. (2012). We now explicitly point to Friston 2018 for readers interested in how representational enhancement fits in the hierarchical predictive coding framework. (p12).

4) There are some inconsistencies in the use of bold type in the text and equations on pages 25 and 26: variables y and r , are sometimes written in bold, and sometimes not.

Thanks very much for spotting these! We have now fixed these typos. (p28)

Reviewer #2

The present work uses an fMRI decoding approach to understand the long debated neural mechanisms underlying the word superiority effect. The experimental design is elegant and the results provide an interesting account of changes in the neural representation of individual letters that are embedded in words as compared to

pseudowords. Overall I found the results compelling and the methods sound. There are two major points that I would like to see addressed in a revision and four additional points that I think merit additional consideration.

Sincerely, Jason Yeatman

We would like to thank the reviewer for his positive comments on, and useful suggestions for improvements of, our work.

First, the VWFA, in fact, responds more to pronounceable pseudowords than real words. Conventionally the VWFA is localized by comparing the response to words against the response to other image categories. So by defining cortex that responds more to words than pronounceable pseudowords it is unclear what region is actually being localized (it has been reported that the anterior fusiform responds more to words than pseudowords but this is still not a conventional localizer). This needs further investigation. Ideally they would use another localizer as critical components of their interpretation rest on this region of the reading network. However, if that is not possible, then more needs to be done to show what is being localized in individual subjects. Are they finding a consistent region in each individual? How does the region they are finding relate to the broader literature? Where is this region positioned anatomically relative to the typical location of the VWFA (the VWFA has a very stereotyped location in the occipitotemporal sulcus)? A general point is that the manuscript would benefit from integrating a bit more of the literature on the VWFA.

We want to thank the reviewer for pointing us to this issue, which highlights a lack of clarity on our side. Critically, the nonword condition did not consist of pronounceable pseudowords, but rather of unpronounceable (orthographically illegal) letter strings. These nonwords were matched in letter frequency (unigram probability), but differed solely (and maximally) in the orthographic sequence (bigram/trigram) probability.

Although other functional localizers exist, simply contrasting familiar words with unpronounceable nonwords (often consonant strings) has been used before to localize VWFA (e.g. Cohen et al. 2002). Responses in VWFA are known to increase monotonically

as a function of word-likeness, both when this is quantified in terms of orthography (Binder et al. 2006) and when quantified in terms of visual features using a visual template model (Kay and Yeatman 2017). Contrasting words with unpronounceable nonwords can be seen as comparing two stimulus categories that are placed relatively far apart on the word-nonword continuum. As such, we believe that our definition of VWFA is in line with VWFA literature and should, a priori, localise the VWFA as it is described in the literature.

To validate, a posteriori, that this was the case, we visually compared the average and individual VWFA ROIs (as defined in the manuscript) to VWFA as defined in two recent and influential studies (Yeatman, Rauschecker, and Wandell 2013; Rauschecker et al. 2012). As a first approximation, we averaged all subject-specific binary VWFA masks and projected this average onto the standardised cortical surface (Figure 1a, below). As can be seen, the mask is centred around a similar occipitotemporal region as typically localized in the literature; see **Fig R1c** (Yeatman, Rauschecker, and Wandell 2013) and **Fig R1d** (Rauschecker et al. 2012). To get a sense of the consistency, we then also projected all individual binary masks onto the standardized cortical surface (**Figure R1b**). This reveals that the ROI was indeed quite consistent across subjects, despite some variability, which is expected for higher-order visual cortex, especially for volumetric analyses. Note that **Fig R1c** and **R1d** are based on a different localizer contrast (word stimuli > phase-scrambled word stimuli) which makes the similarity to our ROIs all the more reassuring.

Importantly, using this definition (words>nonwords) rather than a different one (e.g. words > phase-scrambled words) allowed us to use the main task as a localiser, which meant we had much more data per participant than would have been available using a separate localiser run. We have now further clarified and motivated this decision (P25). Moreover, to avoid confusion, we have clarified our definition of nonwords as unpronounceable letter strings (rather than pseudowords) throughout the text.

Figure R1. Figure 1. Comparison of VVFA ROI definitions. **A)** Average VVFA definition. To obtain the ‘average location’ resulting from our VVFA definition, all individually-defined binary masks were averaged in MNI space and projected onto the average cortical surface, fsaverage. **B)** Individual VVFA masks. To estimate consistency across subjects all individual binary masks were projected on the standard cortical surface. Note that all masks were binary, but on the surface range between 0 and 1 due to interpolation and smoothing (4mm) during projection. **C)** VVFA (red outline with significance blobs) as it was localised by Yeatman et al. (2013). **D)** VVFA as it was defined and functionally localized by Rauschecker et al. (2012)

Second, the theory presented in the introduction rests on the assertion that representations are changing in early visual cortex as opposed to, for example, a change in SNR across much of the brain. Can the authors conduct a searchlight analysis showing correlation and classification accuracy across the cortex (ideally using a similar size searchlight to early visual cortex ROI)? Does this show a peak in early visual cortex (specifically around fovea)? Or can U and N be decoded

throughout much of the brain and is the representation different for words versus pseudowords? If so, how does this affect the theory?

The reviewer raises two important points. First, are we indeed decoding retinotopic visual representations (as our interpretation assumes) or are we relying on much more global information, and are the middle letters decodable throughout much of the brain? Second, is the representational *enhancement* we observe specific to retinotopic visual representations, or does it reflect a change in SNR across much of the brain? The reviewer then helpfully suggests that both points may be addressed by performing a brain-wide searchlight variant of the MVPA analyses.

We agree with the reviewer that these are important points and have performed the suggested analysis. Specifically, we performed classification and pattern correlation analyses with the same procedure as described in the manuscript. However, instead of using a limited number of a priori defined ROIs, we used a spherical searchlight ROI that slid across the brain. A searchlight radius of 6mm was used, yielding an ROI size of ± 170 voxels on average, similar to the 200 voxels in our main ROI.

Below we visualise the group results of both searchlight analyses. To address both points by the reviewer, we use a dual coding scheme. In Figure 2 and Figure 3, the *opacity* conveys the extent to which we can decode the middle letter, while the *colour* conveys the difference in decoding between conditions (word-nonword). From Figure 2-3, two things become clear. First, we almost exclusively see blobs in visual regions, implying that only there decoding was above chance (opacity above 0). This means we are not relying on a global signal that can be decoded throughout much of the brain, but rather on a signal from visual cortex. Second, almost all blobs are red. This means that when there is above-chance decoding, the difference between conditions is almost always positive (i.e., decoding is higher in word compared to nonword context).

Figure R2 Searchlight classification analysis results. We use a dual coding scheme, in which the opacity of the overlay is determined by the average decoding accuracy with respect to chance (averaged over subjects), and the colour indicates the decoding difference (word-nonword) between conditions.

Figure R3. Searchlight pattern correlation analysis. We use a dual coding scheme, in which the opacity of the overlay is determined by the average decoding performance (quantified via pattern information, as in the main pattern correlation analysis) and the colour indicates the difference in pattern correlation between conditions (word minus nonword).

From these figures it is clear that the decoded signal is not global, but specific to (early) visual cortex. However, from the maps it is not easy to see if the effect is foveal or not. This is because the maps got smeared out during averaging. Note that for individual subjects, the searchlight analysis was done in native EPI space (like all MVPA analyses) and the resulting maps were in fact quite sparse. However, variabilities in (functional) anatomy and inevitable imperfections in registration caused the results to be smeared out during averaging.

Therefore, to answer question 2 of the reviewer – i.e. whether letter decoding and representational enhancement peak above the expected (foveal) retinotopic locations – it is

more informative to analyse participants' searchlight results in their native space. As an illustration, we have visualised individual searchlight maps for six participants below. For each participant, searchlight results were registered to their T1 and projected to their native cortical surface using freesurfer. We again use a dual coding scheme in which opacity represents the average decoding performance, and colour represents the difference in decoding (word-nonword). From these maps three things stand out. First, despite some blue and white spots, peaks are mostly red, implying that at locations with the best decoding performance, letter decoding is better for words than nonwords. Second, contrary to the averaged maps, the peak of decoding is very focal. Finally, all foci, despite some individual variability in their exact location, are located in the direction of the posterior (foveal) side of V1/V2, corresponding to retinotopic representations of the centre of the visual field.

Figure R4 Searchlight results on the native cortical surface of 6 representative participants. In both panels, a dual coding scheme is used such that the difference in decoding between conditions is only shown for those locations where the average letter decoding was (well) above chance. **A) Letter classification analysis.** Maps show difference in accuracy between conditions (colour), weighted by average classification accuracy across conditions (opacity). The lower limit of 58% was motivated by the lower accuracy limit that can be considered significantly

greater than chance in a binomial test with 192 samples and an α of 0.05 (i.e. 58.85%). As such, all vertices with visible overlay can be considered as having significantly above chance decoding performance. Maps correspond to left hemispheres of subject 18,28 and 34 (top to bottom). **B) Pattern correlation analysis.** Maps represent difference in correlation-difference between conditions (colour), weighted by average pattern information, i.e. correlation-difference ($\rho_{within} - \rho_{between}$), across conditions. Upper and lower bounds were individually set at a positive number for each subject so only vertices with positive decoding are visible and location of peak decoding peak was emphasised. Brains correspond to right hemispheres of subject 13,24 and 32 (top to bottom). In both panels, only one hemisphere per participant was shown for compactness, since in many participants, the decoding peak (in surface space) was centred at one of the hemispheres. It is unclear if this reflects true hemispheric dominance, or an artefact of the projection due to imperfections in the registration of the T1 and EPI data.

Although these individual maps are striking, visually inspecting a subset of the participants has obvious drawbacks. Therefore, we also approached the question of central/peripheral decoding and enhancement more systematically across all participants, by functionally defining a peripheral visual ROI for each participant. This demonstrated that both letter decoding (irrespective of condition) and representational enhancement (i.e. the difference between word and nonword conditions) were significantly reduced in the peripheral ROI compared to the central ROI. Indeed both decoding and enhancement were so greatly diminished that they were practically absent in the peripheral ROI. For details on this ROI analysis, see the second response to Reviewer #3, who raised a similar question.

We have now added our analyses of spatial and retinotopic specificity to the results (p8), and the supplement (p S6-S9), and added corresponding files and code to the repository.

Minor comments

It would be helpful for me to see the bold amplitudes in the main text rather than just the supplement and to show the mean difference between conditions (in addition to the violin plots). The violin plots make it clear that there are large inter-subject differences in BOLD amplitude but it is hard for me to see how consistent the mean difference between conditions is. This is a very minor point.

Below we have included the difference plot, both for the main early visual cortex ROI (Figure R5) and for the other visual ROIs (Figure R6). Figure R5 was also added to the univariate result in the Supplement (see Figure S4b). Regarding its position in the text: we consider the univariate result to be a control analysis (albeit an important one) and not a key finding on its own, or of critical importance to understand the rest of the manuscript. Adding it as a subpanel to Figure 3 or 4 seems out of place, and adding it as a separate figure seems arbitrary, because it is only one of multiple important control analyses. For this reason, we prefer to keep it in the supplement.

Figure R5 No univariate difference in early visual cortex. **A.** Mean BOLD amplitude estimates for word and nonword conditions. **B.** Amplitude difference between conditions for each participant.

Figure R6 Univariate differences for all ROIs.

The whole-brain results shown in figure 4c appear to use a drawing of the brain. Could the authors switch to a more conventional approach such as a cortical surface derived from freesurfer?

The whole-brain results of Figure 4c are displayed using a so called 'glass brain plot'. This transparent rendering of the brain is conventional in neuroimaging (popularised by the MATLAB software package SPM). However, we did not use the well-known SPM version,

but the version from the python package nilearn, which is newer and therefore not yet as widely used. We would prefer to keep the glass brain plot in Figure 4c because it quickly and directly reveals all potential clusters throughout the brain. As such, it nicely aligns with the purpose of the exploratory whole-brain analysis of 4c – that is, to exploratorily search throughout the entire brain for other potential clusters showing the same effect we established in our predefined ROIs. However, we contend that the minimalistic character of the plot omits a considerable amount of anatomical and statistical detail. We have therefore now added a volumetric slice by slice rendering of the same results to the supplement (Fig S11, page supplement 11). Importantly, this volumetric rendering was not discretely thresholded (but rather opacity-weighted by statistical significance) so it also gives a richer picture of the results. We prefer this volumetric visualisation over a surface plot, because a surface plot can give a misleading sense of precision by suggesting that we did the analysis in surface space rather than (like all MVPA analyses) in volume space, and projecting to the standard surface only after performing the alignment and averaging in volume space.

Rather than using an affine alignment for the group results, cortex based alignment would be preferable. Particularly in ventral temporal cortex, inter-subject variability is accounted for much better by cortex based alignment than volume based alignment (and a simple affine will leave a lot of anatomical variability). Work by Frost and Goebel as well as work from Weiner and Grill-Spector has shown the large benefits of cortex based alignment within ventral temporal cortex.

We have chosen for linear registration because of previous mixed experience with FSL's nonlinear registration. While we are aware that more sophisticated registration methods are available, we want to emphasise that group alignment only affects the additional, exploratory analysis that is displayed in Figure 4c. All primary, predefined analyses were done in native space with functionally defined subject-specific masks, and are hence unaffected by using a different registration method. We have now further emphasized the exploratory character of the whole-brain analysis in 4c and discuss this limitation in the text (p 21). We want to thank the reviewer for the suggestion and the referenced work, and will consider using cortex based alignment for future studies, especially for ventral temporal cortex.

In the main experiment the words are embedded in noise. What is the significance of

this decision and how much do the results depend on the stimulus being embedded in noise?

We have two reasons for using noise, one principled and one more pragmatic. The first is that, while according to theoretical models there is always *some* enhancement, the effect is much larger under visually challenging conditions (e.g. under visual noise). This is because when stimuli are presented under optimal conditions, the representational strength of single letters or nonwords will be already near ceiling (in our definition near the asymptotic value of 1), leaving little room for contextual enhancement – a neural/cognitive ceiling effect.

The second reason we used noise also boils down to avoiding a ceiling effect – but one in the measurement. In particular, even without the neural/cognitive ceiling effect, it could be that without adding visual noise, fMRI decoding might be close to ceiling for nonwords, making it much harder to pick up on potential increases in decoding performance by word contexts. In other words, although this is an empirical question, we believe that noise could be necessary to *observe* the effect. But this does not mean that without noise the effect would not occur – simply that it would be smaller and/or more difficult to detect.

Of note, adding noise is by no means unique to our experimental paradigm. Almost all behavioural studies on word superiority try to create challenging viewing conditions, by using brief presentation durations combined with masking, in order to robustly study contextual enhancement effects.

Small notes:

p3: what is a target? is a spelling task an orthographic task?

Changed orthographic to ‘spelling’, because for the nonword targets the ‘correct spelling’ is not in line with ‘orthography’ of Dutch (because they are orthographically illegal nonwords).

p3: behavioral result: what are the effect sizes, in units of RT (ms) and accuracy?

Added effect sizes: “Participants were faster (median RT difference: -29.2 ms; Wilcoxon signed rank, $T_{34}=40$, $p=1.07\times 10^{-5}$, $r = 0.87$) but not significantly more accurate (median accuracy difference: 1.62%; *t*-test, $t_{34} = 1.70$, $p = 0.098$, $d = 0.29$) for word compared to nonword targets.” (P.3).

p5, line 9: what does it mean that the SNR was identical? SNR of what?

The amount of noise added to the stimuli in the simulation (and the experiment) was identical for words and nonwords. As such words and nonwords were objectively (in terms of stimulus energy) equally perceptible. We have changed the phrasing (see p5).

To what extent is the ROI stimulated by letters to the side and does that matter?

We do not expect that the ROI is purely sensitive to the middle letter. However, the extent to which they also respond to letters on the side is impossible to say since we did not perform an experiment during which there was only stimulation at the side and not in the middle. For our purposes, what matters is that at least the presence of letters on the side (and visual noise) did not completely change the evoked activity pattern, because the letter classifier trained on single letters (without noise) could generalise well to full words (with noise).

Reviewer #3

The authors report an investigation into the neural bases of contextual effects in perception. Specifically, they test an account of the word-superiority effect (better representation of individual letters in context of a real world vs non-word) that is based on top-down influences on perception. This is distinguished from a post-perceptual, decisional account of the phenomena. Participants viewed words and nonwords, each with one of two possible central letters. fMRI classification measures showed that discrimination of the central letter was improved, for a range of early visual-cortex ROIs, in words vs non-words. Further, a PPI-style analysis identified candidate neural sources of hypothesised top-down effects in visual word form area, pMTG, and IFG – all regions with good a priori reasons to be implicated in word processing. Control analyses excluded influences of main-effect (univariate) differences on the classification results, and differences in eye movements. The authors conclude that contextual effects on letter perception must at least in part be due to top-down facilitation of early perceptual processes.

The manuscript is clear and thorough, and addresses a problem of both long-standing and current interest. Numerous strengths are in evidence, for example: a relatively large N compared to typical studies; the use of independent localisers and hypothesis-driven regions of interest (and the latter tested over a range of voxel numbers); the use of a formal modelling approach to establish connection between

theory and data; careful balancing of the two stimulus classes; and testing eye movements to rule out potential confounds in attention to words vs nonwords.

We appreciate the positive and thorough assessment of our manuscript.

The main limitation in my view is that the authors' logic implicitly assumes that evidence of representational quality effects in retinotopic visual cortex are evidence only for a top-down effect on "purely" perceptual representations. This logic should be further exposed – that is, the authors should elaborate on the argument that other non-perceptual (e.g. decision-related) effects could not in principle manifest as influences on early visual cortical areas. Here we must keep in mind that obviously BOLD activity captures the full time-course of activity elicited by the word/non-word stimuli and is not differentially sensitive to earlier vs later waves of neural activity.

We agree with the reviewer that our inferential logic merits some additional consideration, and want to thank them for encouraging us to be conceptually more careful.

Specifically, we agree that fMRI cannot differentiate between earlier vs later activity. Hence it is possible that the observed effects arise late, perhaps even much later than what is typically considered 'perceptual' (e.g. > 500 ms). We do not believe we can rule out this possibility *in principle*. However, what we can do is make a plausibility argument for why we believe our interpretation (perceptual enhancement) is the most likely.

First, several non-perceptual explanations related to confounds (such as BOLD amplitude or eye movements) were rejected based on additional analyses (as commended by the reviewer). Second, psychophysical studies have shown that letters are genuinely more easily *recognised* (as opposed to being merely more easily reported or remembered) in words than nonwords, making the idea of enhanced perceptual representations plausible (e.g. Lupyan 2017). Third, qualifying representational effects as perceptual or non-perceptual based on their latency and reliance on feedback of higher-order regions raises the thorny question of what, exactly, counts as perceptual and what does not. Surely effects do not have to be immediate to be perceptual, as this would exclude all cortical feedback effects from the realm of vision. Indeed, in the simulations we present the effects are not immediate and take tens of iterations to develop. One might feel inclined to label only early feedback effects as perceptual, while labelling effects after a certain moment differently, e.g.

as iconic memory. Exactly defining this border, however, does not seem trivial as the transition may well be gradual. One possibility to examine this issue more deeply is running a study with high temporal resolution recording method (like MEG or ECoG), combine long inter stimulus intervals with relatively brief stimulus presentation durations, and only label representational effects *before* offset as perceptual.

Although this may be an interesting direction for future work, our current goals were much humbler: we merely tried to test whether we could – as top-down models of visual word recognition predict – find evidence for a neural feedback effect on sensory representations. We indeed found evidence for such an effect. The fact that we do not know its latency does not undermine this finding, because a late feedback effect is still a feedback effect.

More generally, it might be useful to distinguish between *representational enhancement* (i.e. top-down enhancement of *sensory representations*) and *perceptual enhancement* (i.e. top-down enhancement of *perception*). Given that we performed numerous controls for non-representational confounds (differences in amplitude, ROI-definition contingencies, eye-movements) and established that the MVPA results were retinotopically specific, we believe our results provide strong evidence for *representational enhancement*. This is significant, because it matched our prediction: if word superiority reflects *perceptual enhancement* of letters (as influential top-down models assume), then it should be accompanied by *representational enhancement* in visual cortex – which is precisely what we found.

But the reviewer is correct to point out that there is a logical implication here, which cannot be simply inverted: while perceptual enhancement *necessarily* implies representational enhancement, representational enhancement does not *necessarily* imply perceptual enhancement, as it might constitute a post-perceptual enhancement of sensory representations, like in the case of enhanced post-perceptual iconic memory encoding.

We believe that in our case, the perceptual interpretation is most likely, but we concede a non-perceptual interpretation cannot be definitively ruled out. This might be an avenue for future studies, that could either try to probe perception more directly (e.g. using an objective behavioural measure of perception in addition to a neural measure of representation) or by using a high temporal resolution recording method in combination with a temporal criterion to arbitrate between ‘perceptual’ and ‘post-perceptual’ representational enhancement.

We have now improved the discussion to include the distinction between representational enhancement and perceptual enhancement and discuss potential future studies to tease the two apart (p15). Moreover, we have toned down some of the claims throughout the manuscript, and adjusted the terminology in all cases where *perceptual enhancement* and *representational enhancement* were used interchangeably.

Second, the authors note that “enhancement was consistently found in multiple visual areas”. As a complement, to further demonstrate the specificity of the context effect, it would have been useful to identify a control visual region(s) where differential effects of context would not be expected. For example, given the strong retinotopy of V1, it should be possible to distinguish foveal voxels that are presumably the main target of the central U and N stimuli, from more peri-foveal or peripheral voxels, that presumably would not be so strongly influenced by the interaction of the central item with the word/non-word context. A demonstration of such retinotopic specificity would strengthen the authors’ claims that what is being influenced by context is indeed the specific, early perceptual representation of the critical items.

We agree with the reviewer that this is an important point and want to thank her/him for suggesting an additional analysis to address it. In fact, reviewer #2 raised a closely related point and in response we performed a searchlight variant of the MVPA analyses. Here, we use the resulting searchlight maps (containing classification and pattern correlation results for each voxel in a participant’s native EPI space) to perform the suggested analysis.

As suggested, we leverage the strong retinotopy of V1 to test for enhancement in peri-foveal or peripheral voxels. In each participant, we used results from the letter localiser and main experiment to identify V1 voxels that did not respond to the single letters (presented at fixation), but did respond vigorously to stimulation in the main experiment (which covered a large part of the visual field). Specifically, we compared the contrast stimulation>baseline in the localiser with that same contrast in the main experiment, and looked at voxels that were both in the lower half of the first contrast and in the higher half of the second contrast in terms of Z-statistic. This conjunction resulted in ROIs of $180\pm$ voxels on average, for which it was confirmed that the Z-statistic for the main experiment was sizable (4.3 on average) while that of the letter>baseline comparison was close to zero (0.8 on average).

As can be seen in Figure 7, letter decoding is now barely above chance. Moreover, the enhancement effect is greatly reduced: it is absent for the classification analysis (Wilcoxon sign-rank, $T = 290$, $p = 0.9$, $r = 0.025$) and only slightly significant for the pattern correlation analysis (paired t-test, $t_{34} = 2.66$, $p = 0.02$, $d = 0.456$). When the enhancement effect is compared to the effect observed in the foveal V1 ROI (as defined in the manuscript), we find the effect is significantly smaller, both for classification (paired t-test, $t_{34} = 2.57$, $p = 0.015$) and for the pattern correlation analysis (paired t-test, $t_{34} = 2.92$, $p = 6.31 \times 10^{-3}$). After performing this analysis within V1 (as the reviewer requested) we repeated the same procedure for early visual cortex (i.e. the conjunction of V1-V2). Again we found reduced overall letter decoding, both for the classification analysis (paired t-test, $t_{34} = 18.49$, $p = 5.52 \times 10^{-19}$, $d = 3.17$) and pattern correlation analysis (paired t-test, $t_{34} = 8.86$, $p = 3.02 \times 10^{-10}$, $d = 1.52$). Moreover, we again found a reduction of the enhancement effect, again both for the classification analysis (paired t-test, $t_{34} = 2.44$, $p = 0.02$, $d = 0.42$) and pattern correlation analysis (paired t-test, $t_{34} = 3.21$, $p = 2.90 \times 10^{-3}$, $d = 0.55$).

Together, these results show that the MVPA results are indeed are retinotopically specific, providing further evidence that the MVPA results indeed reflect sensory representations, as expressed in BOLD activity. We have included these analyses to the manuscript (see P8, Figure S9 and Supplemental result 1).

Figure R7. Reduced letter decoding performance and representational enhancement in V1 periphery.

Specific points:

It is not clear from the Methods whether the U/N localiser took place before or after the main experiment. If it took place first, this could have had the effect of biasing participants' attention towards the central U/N in the main task. While that in itself would not explain the current findings, it could have the effect of unnaturally increasing the strength of that central letter's neural representation in a way that would not be the case in natural word reading.

The U/N localiser took place after the main experiment. We now make this explicit in the methods section (P20):

Page 13 A further complication to this picture, not discussed by the authors, is found in evidence for "contextual suppression" e.g. Suzuki and Cavanagh, 1995 found that whole-face configuration can impede perceptual access to individual features. Consistent context is not always facilitatory – and so in some cases early visual representations of individual elements should be weaker within a meaningful context, rather than stronger.

This is an interesting observation, of which we were not aware, and which certainly further complicates the picture. We have now included the fact that context, paradoxically, can also occasionally have detrimental effects on the identifiability of parts in our discussion (P14)

References

- Binder, Jeffrey R., David A. Medler, Chris F. Westbury, Einat Liebenthal, and Lori Buchanan. 2006. "Tuning of the Human Left Fusiform Gyrus to Sublexical Orthographic Structure." *NeuroImage* 33 (2): 739–48. <https://doi.org/10.1016/j.neuroimage.2006.06.053>.
- Cohen, Laurent, Stéphane Lehericy, Florence Chochon, Cathy Lemer, Sophie Rivaud, and Stanislas Dehaene. 2002. "Language-specific Tuning of Visual Cortex? Functional Properties of the Visual Word Form Area." *Brain* 125 (5): 1054–69. <https://doi.org/10.1093/brain/awf094>.
- Friston, Karl. 2018. "Does Predictive Coding Have a Future?" *Nature Neuroscience* 21 (8): 1019–21. <https://doi.org/10.1038/s41593-018-0200-7>.

- Kay, Kendrick N, and Jason D Yeatman. 2017. "Bottom-up and Top-down Computations in Word- and Face-Selective Cortex." Edited by Joshua I Gold. *ELife* 6 (February): e22341. <https://doi.org/10.7554/eLife.22341>.
- Kok, Peter, Janneke F. M. Jehee, and Floris P. de Lange. 2012. "Less Is More: Expectation Sharpens Representations in the Primary Visual Cortex." *Neuron* 75 (2): 265–70. <https://doi.org/10.1016/j.neuron.2012.04.034>.
- Lupyan, Gary. 2017. "Objective Effects of Knowledge on Visual Perception." *Journal of Experimental Psychology. Human Perception and Performance* 43 (4): 794–806. <https://doi.org/10.1037/xhp0000343>.
- Rauschecker, Andreas M., Reno F. Bowen, Josef Parvizi, and Brian A. Wandell. 2012. "Position Sensitivity in the Visual Word Form Area." *Proceedings of the National Academy of Sciences of the United States of America* 109 (24): E1568-1577. <https://doi.org/10.1073/pnas.1121304109>.
- Yeatman, Jason D., Andreas M. Rauschecker, and Brian A. Wandell. 2013. "Anatomy of the Visual Word Form Area: Adjacent Cortical Circuits and Long-Range White Matter Connections." *Brain and Language* 125 (2): 146–55. <https://doi.org/10.1016/j.bandl.2012.04.010>.

Reviewers' Comments:

Reviewer #1:

Remarks to the Author:

The revised manuscript adequately addresses my initial concerns, except for issue number 3. In an attempt to address this concern the authors have added a reference to Friston 2018. However, Friston 2018 is primarily concerned with the role of cortical feedback in mediating precision-weighting: a topic that is entirely orthogonal to the subject of the current manuscript. The addition of this reference makes the relationship between the current results and predictive coding less clear, not clearer. While I think this unfortunate, it is not an issue of sufficient concern to prevent publication.

Reviewer #2:

Remarks to the Author:

The authors have done a thorough job responding to my comments

Reviewer #3:

Remarks to the Author:

The authors have thoroughly addressed the critiques I raised in the first review, which primarily related to 1) clarifying the logic of the authors' claims with relation to perceptual vs representational enhancement effects and 2) better demonstrating that their observed effects are retinotopically specific, as would be expected based on their framework.

I have no further comments or critiques.

Reviewer #1 (Remarks to the Author):

The revised manuscript adequately addresses my initial concerns, except for issue number 3. In an attempt to address this concern the authors have added a reference to Friston 2018. However, Friston 2018 is primarily concerned with the role of cortical feedback in mediating precision-weighting: a topic that is entirely orthogonal to the subject of the current manuscript. The addition of this reference makes the relationship between the current results and predictive coding less clear, not clearer. While I think this unfortunate, it is not an issue of sufficient concern to prevent publication.

On second consideration we agree with the reviewer that adding a reference to Friston (2018) might not be helpful after all. It is true that Friston (2018) is primarily concerned with second-order expectations (i.e. uncertainty/precision) which we do not explicitly manipulate or model. We have therefore removed the reference.

However, we stand by our main point, namely that top-down representational sharpening is not inconsistent with predictive coding formulations. In fact, representational sharpening is already discussed explicitly in the seminal paper by Friston (2005). There, Friston (2005, p.829) discusses the tension between sharpening and dampening/cancelling of representations, and writes that the tension is resolved by postulating that both processes occur, albeit in distinct subpopulations, error units and representation units.

Elaborating on this in detail in the main text seems like a distraction since we only mention predictive coding very briefly in our discussion (for reasons we have outlined in the first reply to the reviewers). However we have now added the reference to Friston (2005) for readers curious about representational sharpening within hierarchical predictive coding theories.

References

Friston, K. (2005). A theory of cortical responses. *Philosophical transactions of the Royal Society B: Biological sciences*, 360(1456), 815-836.